# Beyond Dual Representations: Collaborative Learning for Semi-Supervised LiDAR Semantic Segmentation

## Abstract

Annotating large-scale LiDAR point clouds for 3D semantic segmentation is costly and time-consuming, motivating the use of semi-supervised learning (SemiSL). Standard SemiSL methods typically rely on a single LiDAR representation in a two-step framework, where consistency between identical models is enforced under input perturbations. However, these approaches treat pseudo-labels from a single network as fully reliable, which reinforces architectural biases and propagates errors during distillation, ultimately limiting student performance. Recent dual-representation methods alleviate this issue but still remain constrained by the limitation of two-step design. We introduce *CoLLiS*, a novel framework that leverages **Co**llaborative **L**earning for **Li**DAR **S**emi-supervised segmentation. Unlike prior paradigms, *CoLLiS* trains multiple representations collaboratively in a single step by treating them as coequal students. Cross-representation distillation is adaptively balanced by monitoring inter-student disparities to mitigate confirmation bias and improves robustness. Extensive experiments on three public benchmarks show that *CoLLiS* consistently enhances the performance of all participating models and achieves superior results compared to state-of-the-art LiDAR SemiSL methods. The code will be released upon acceptance.

## 1 Introduction

The robustness of LiDAR sensors in environmental perception has propelled their widespread adoption for 3D scene understanding in autonomous driving. The inherent geometric challenges of LiDAR data, such as sparsity and viewpoint distortions, have motivated extensive research into diverse input representations for semantic segmentation. These approaches predominantly leverage range-view images Cortinhal et al. (2020); Milioto et al. (2019), voxel grids Zhu et al. (2021); Choy et al. (2019) and polar images Zhang et al. (2020). To combine complementary advantages of different LiDAR geometry in perception, prior works progressively focus on fusing different LiDAR representations Xu et al. (2021); Hou et al. (2022).

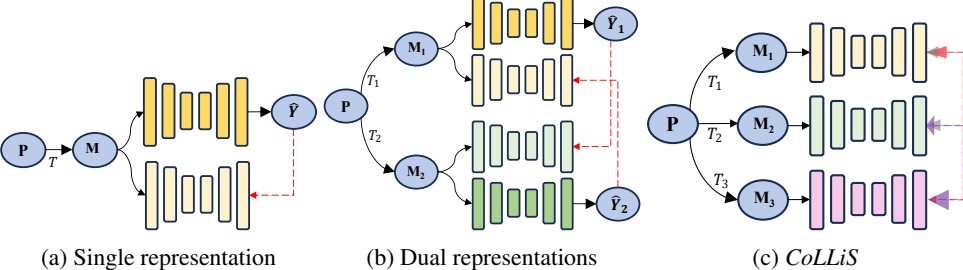

(a) Single representation   (b) Dual representations   (c) *CoLLiS*

Figure 1: Point cloud $\mathbf{P}$ is transformed into multiple LiDAR representations $M_i$ using their respective transformations $T_i$. (a) Prior works Kong et al. (2023c); Chen et al. (2021b); Li et al. (2023) involves two decoupling steps: pseudo-labeling and distillation, with identical networks employed in both. (b) Recent advances Liu et al. (2025) extend the framework by incorporating dual LiDAR representations and optimizing networks via cross distillation. (c) In contrast, *CoLLiS* leverages multiple representations within a single-step framework to streamline the process and employs adaptive mutual distillation to enhance generalization.

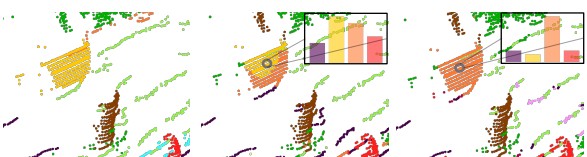

Figure 2: Confirmation bias: distillation from a single model can over-fit to its own errors. From left to right: ground-truth labels, predictions at $t_1$, and predictions at $t_2$ ($t_1 < t_2$).

Despite these advancements, existing methods primarily rely on fully supervised training, requiring vast amounts of fine-grained annotated data to achieve baseline accuracy. Annotating large-scale LiDAR datasets, however, is prohibitively time-consuming and labor-intensive. Limitations have driven the exploration into semi-supervised learning (SemiSL) for LiDAR semantic segmentation, where models train on limited labeled data alongside abundant unlabeled samples. Early LiDAR SemiSL works adopt consistency regularization through input perturbations using two-step frameworks Kong et al. (2023c); Li et al. (2023); Unal et al. (2022). Recent works further extend these strategies to dual representations Liu et al. (2025), which shows significant improvements by training with dual LiDAR representations. In particular, the success stems from utilizing the complementary information from different LiDAR representations, thereby improving generalization mainly by leveraging invariants Rath & Condurache (2020) effectively. For instance, range-based view is generally dense but contains distorted artifacts due to projection, while voxel-based view is regular but may struggle to capture discriminative features in sparse regions Xu et al. (2021).

However, existing SemiSL methods remain vulnerable to **confirmation bias** Arazo et al. (2020), where erroneous pseudo-labels are reinforced during training, especially under high label scarcity. This issue is exacerbated in LiDAR segmentation due to the long-tail distribution of objects and the architectural biases of individual networks. Dual-representation approaches alleviate the problem to some extent by providing complementary perspectives Coors et al. (2018), but they are still limited by their two-step design with unidirectional distillation from a single network. As a result, the model risks over-fitting to noisy predictions and failing to generalize.

To address this challenge, we propose *CoLLiS*, a novel collaborative framework for semi-supervised LiDAR semantic segmentation. Unlike conventional two-step pipelines that rely on pseudo-labels from a single model, *CoLLiS* treats multiple LiDAR representations as coequal student models that learn collaboratively in a single step. Through online distillation over multiple representations, our framework aggregates complementary information, benefits from the inductive bias of individual models, and reduces dependence on any single network. Thus, it effectively mitigates confirmation bias. Moreover, the streamlined single-step design improves efficiency when scaling to multiple representations.

Our contributions are as follows:

1. We introduce a consensus-driven augmentation mechanism that dynamically adjusts augmentation strength based on peer agreement, thereby enhancing generalization in SemiSL. (Sec. 3.2.1)
2. We propose an adaptive pseudo-labeling and distillation strategy that balances online knowledge transfer across multiple models by accounting for inter-student disparities (Sec. 3.2.2).
3. Beyond evaluating the collaborative learning framework, we conduct a post-training ensemble analysis, an aspect largely unexplored in LiDAR SemiSL, and demonstrate its utility for further improving performance (Sec. 3.2.4).

## 2 RELATED WORKS

**LiDAR segmentation and SemiSL** LiDAR semantic segmentation methods differ by input representation: some process raw point clouds Wu et al. (2024); Puy et al. (2023), while others impose structure through range/polar projections Zhao et al. (2021b); Zhang et al. (2020) or voxel grids Zhu et al. (2021); Choy et al. (2019). Multi-view fusion leverages complementary cues Hou et al. (2022); Xu et al. (2021), but all require costly large-scale annotations. Semi-supervised learning (SemiSL) reduces this burden by exploiting unlabeled data. Recent works adopt contrastive learning Li et al. (2023); Li & Dong (2024) or consistency with LiDAR-specific perturbations Kong et al. (2023c).

Figure 3: Overview of *CoLLiS*. ① The labeled dataset $D_l$ is repeated to match the size of the unlabeled dataset $D_u$. A batch from each is sampled and optionally mixed using a random mixing strategy with non-fixed probability. ② The sampled point clouds are transformed into multiple LiDAR representations. ③ Each model is trained with a composite loss consisting of a labeled loss ($L_l$), a regularization term ($L_{reg}$), and an unlabeled loss ($L_u$). Pseudo-labels are generated online using confidence-based modeling.

IT2 Liu et al. (2025) improves performance by enforcing cross-representation consistency between dual views, highlighting the benefit of complementary information. Yet most methods rely on two-step pipelines, limited to one or two representations, and still suffer from confirmation bias due to distillation from a single model. We address this with a one-step collaborative framework that integrates multiple representations for more scalable and robust LiDAR SemiSL.

**Collaborative learning (CoL)** eliminates the need for explicit pseudo-labeling by enabling peer networks to learn from each other. DML Zhang et al. (2018) pioneered reciprocal supervision between two networks, while KDCL Guo et al. (2020) and ONE Zhu et al. (2018) introduced ensemble-based pseudo-teachers. More recent studies Liu et al. (2022); Zhu et al. (2023) extend this idea to heterogeneous architectures, such as CNNs and Vision Transformers Dosovitskiy et al. (2021), demonstrating enhanced generalization through collaboration among heterogeneous models. Despite these successes, CoL remains underexplored in semi-supervised settings and largely absent in the LiDAR domain, where sparse geometry and noisy pseudo-labels pose unique challenges.

## 3 CoLLiS

### 3.1 PRELIMINARIES

We consider a LiDAR point cloud with $N$ points $\mathbf{P} = p_i \mid p_i = (x, y, z, I)_i$, where $(x, y, z)$ are coordinates and $I$ is intensity. Training data consists of a labeled set $D_l = (\mathbf{P}_j^l, \mathbf{Y}_j^l)$ with one-hot labels $\mathbf{Y}_j^l \in \mathbb{R}^{N \times K}$ for $K$ classes, and an unlabeled set $D_u = \mathbf{P}_j^u$, with $|D_l| \leq |D_u|$. To impose geometric priors, $\mathbf{P}$ is typically transformed into structured forms such as range images $\mathbf{R} \in \mathbb{R}^{U \times V \times C_r}$ via spherical projection Milioto et al. (2019) or voxel grids $\mathbf{V} \in \mathbb{R}^{H \times W \times L \times C_v}$ via discretization Zhu et al. (2021). Since these representations differ in structure, we use the point cloud as an intermediary to define cross-representation mappings, i.e., range-to-voxel and voxel-to-range transformations can be expressed as $T_{r \to v} = T_{p \to r}^{-1} \circ T_{p \to v}$ and $T_{v \to r} = T_{p \to v}^{-1} \circ T_{p \to r}$. These mappings are essential for enabling effective distillation across different representations.

### 3.2 FRAMEWORK DETAILS

#### 3.2.1 CONSENSUS-DRIVEN AUGMENTATION

In semi-supervised learning, limited annotations restrict generalization and undermine pseudo-label quality. Single-step paradigms apply input augmentations only once per iteration, and requires both diverse training data to improve generalization and reliable pseudo-labels to guide distillation. When augmentation intensity is fixed, they become prone to under- or over-fitting Aliferis & Simon (2024).

A naive solution is to adjust the probability using curriculum learning (CL) Bengio et al. (2009), where augmentation strength gradually increases with training progression. However, CL requires sensitive hyperparameter tuning and enforces a rigid schedule, which is suboptimal. To address this, we propose a consensus-driven augmentation (CDA) mechanism that automatically adjusts the mixing probability ($q_m$) based on inter-student consistency. Our focus is on controlling augmentation intensity dynamically rather than designing new mixing methods. We detail this mechanism below.

We first compute a fraction $a_n$ of predictions that are consistent across students over a step size $N$:

$$a_n = \frac{\sum_{i,j\in\{1,2,3\}}^{i\neq j} \mathbb{I}(\hat{Y}_{n,s_i} = \hat{Y}_{n,s_j})}{\sum \mathbb{I}(\hat{Y}_n)}, \tag{1}$$

where $\mathbb{I}(\cdot)$ is the indicator function. The ratio acts as a measure of data complexity: when students agree, the sample is likely easy to learn and can benefit from stronger augmentations to increase diversity. Conversely, when students disagree, the sample is inherently harder, so weaker augmentations are applied to avoid introducing additional noise. Next we transform these ratios based on whether the point clouds have been mixed:

$$g(x) = \begin{cases} a - 1, & \text{if mixed.} \\ a, & \text{otherwise.} \end{cases} \tag{2}$$

Finally, the mixing probability is updated iteratively based on the past probability:

$$q_{m,t} = q_{m,t-1} * (1 + \sum_{n=1}^{N} g(a_n)), \tag{3}$$

In practice, we observe that CDA consistently outperforms CL in both accuracy and robustness (see Sec. 4.3 for supporting experiments). To further enhance scene diversity, we integrate multiple mixing strategies from prior works: LaserMix Kong et al. (2023c), PolarMix Xiao et al. (2022), and Sub-cloud Shuffling Yang & Condurache (2025), with one strategy randomly selected at each iteration during training (see Appendix B.5 for further details).

### 3.2.2 ON-THE-FLY PSEUDO-LABELING

In *CoLLiS*, each student treats all other participants as potential sources of pseudo-labels but incorporates their knowledge selectively. This selection is guided by two factors: *Absolute Reliability (AR)* and *Relative Reliability (RR)*. AR quantifies a model's intrinsic confidence as training progresses, while RR calibrates reliability by comparing peers. Together, these metrics enhance training robustness by down-weighting unreliable information and balancing knowledge transfer among students of varying strengths.

Motivated by empirical findings that pseudo-label reliability improves as training proceeds Wang et al. (2022), we model AR (denoted as $\beta$) as a linear function of the training epoch and use it to compute the weight of the unlabeled loss ($\lambda_u$):

$$\beta(e) = \frac{e}{E_{\max}}, \quad \lambda_u = \lambda_0(1 - \beta) + \beta, \tag{4}$$

where $e$ is the current epoch, $E_{\max}$ is the total number of training epochs, and $\lambda_0$ is the initial loss weight.

To establish the relative reliability ($RR$), we quantify pairwise prediction confidence, denoted as $\gamma$. Given predictions $P_{s_1}$ and $P_{s_2}$ from two students, we first unify their output dimensions by mapping all representations back to the original point cloud format. We then perform point-wise comparisons and count the number of instances where one branch's confidence exceeds the other's. Formally, the relative reliability of student $s_1$ with respect to student $s_2$ is defined as:

$$\gamma_{s_1\rightarrow s_2} = \frac{N_{s_1\rightarrow s_2}}{N_{s_2\rightarrow s_1}}, \text{ where } N_{s_i\rightarrow s_j} = \sum \mathbb{I}(c(P_{s_i}) > c(P_{s_j})) \tag{5}$$

$c(\cdot)$ is the confidence of prediction and $N_{s_i\rightarrow s_j}$ is the count of points for which confidence of student $i$ exceeds that of student $j$. $\mathbb{I}(\cdot)$ denotes the indicator function. We adopt confidence counts as they better capture discriminative differences between models. In contrast, measures like average confidence tend to over-smooth variations and fails to reflect relative reliability. Intuitively, a model is considered more reliable if it produces higher-confidence predictions more frequently across points.

Next, the threshold value $\delta$ and the pseudo-labels are obtained by Eq. 6 and Eq. 7:

$$\delta(t)_{s_1\rightarrow s_2} = min(\delta_0, \delta_0 * ((1 - \beta) * \frac{1}{\gamma_{s_1\rightarrow s_2}}) \tag{6}$$

$$\hat{Y}_{s_1\rightarrow s_2} = \{argmax(P_{s_1})|c(P_{s_1}) > \delta(t)_{s_1\rightarrow s_2}\}, \tag{7}$$

where $\delta_0$ is the predefined maximum and $\hat{Y}_{s_i \to s_j}$ are pseudo-labels from student $i$ to $j$. Specifically, high absolute and relative reliability will result in the lower threshold value, leading to more tolerant pseudo-labeling and increase in distillation power. All filtered pseudo-labels are subsequently mapped back to model-specific representations to preserve geometric consistency.

### 3.2.3 Loss function

The overall loss function combines following three components:

**Labeled loss** The labeled loss is computed using the ground-truth annotations:

$$L_l = \mathcal{L}(P_s, Y), \tag{8}$$

where $P_s$ denotes the prediction of student $s$ and $Y$ is the ground-truth label. Following prior work Liu et al. (2025); Kong et al. (2023c), the loss function $\mathcal{L}$ is defined as the sum of cross-entropy loss and Lovász loss Berman et al. (2018).

**Unlabeled loss** In collaborative training, confirmation bias can be amplified when multiple representations produce inconsistent pseudo-labels, as naive distillation may over-rely on unreliable predictions. To mitigate this, we assign adaptive weights that regulate the relative influence of each student during distillation. The weights are derived from confidence counts (Sec. 3.2.2), which measure how often one student is more confident than another. For knowledge transfer from students $s_1$ and $s_2$ to a third student $s_3$, the weights are:

$$\omega_{s_1 \to s_3} = \frac{N_{s_1 \to s_2}}{N_{s_2 \to s_1} + N_{s_1 \to s_2}}, \quad \omega_{s_2 \to s_3} = 1 - \omega_{s_1 \to s_3}. \tag{9}$$

These weights help to softly resolve conflicts in predictions while preventing over-reliance on a single knowledge source.

Finally, the unlabeled loss for each student (student 1 as an example) is computed as:

$$L_{u,s_1} = \lambda_u \cdot \sum_{s_i \neq s_1} \omega_{s_i \to s_1} \cdot \mathcal{L}(P_{s_1}, \hat{Y}_{s_i \to s_1}) \tag{10}$$

where $\mathcal{L}$ is the same loss function as in $L_l$.

**Regularization loss** Because *CoLLiS* relies on prediction confidence for pseudo-labeling and distillation, we further regularize against overconfidence using Zou et al. (2019):

$$L_{reg} = -\lambda_{reg} \sum_{k=1}^{K} \frac{1}{K} \log P(k), \tag{11}$$

where $K$ is the number of classes, $P(k)$ the softmax probability, and $\lambda_{reg} = 0.1$. This KL divergence to a uniform distribution smooths predictions and prevents overconfidence.

### 3.2.4 Post-training ensemble

In collaborative learning frameworks such as *CoLLiS*, post-training ensemble provides a simple yet effective way to consolidate knowledge from multiple student models. Since the ensemble is applied to well-trained models, we adopt a straightforward strategy that selects the prediction from the student with the highest confidence:

$$s^* = \underset{s \in 1,...,S}{\arg\max}, c(P_s), \quad P_{ensemble} = P_{s^*}. \tag{12}$$

## 4 Experiments & Discussion

### 4.1 Experimental setup

We evaluate *CoLLiS* on three LiDAR segmentation benchmarks: nuScenes Fong et al. (2022), SemanticKITTI Behley et al. (2019) and ScribbleKITTI Unal et al. (2022). For all three datasets, we

follow the settings of previous works Kong et al. (2023c); Liu et al. (2025): uniformly sampling 1%, 10%, 20% and 50% labeled data for training and the rest of dataset as unlabeled set. In the default setting of *CoLLiS*, we employ FRNet Xu et al. (2023) to process frustum-range-view representation, PolarNet Zhang et al. (2020) for bird's-eye-view images, and Cylinder3D Zhu et al. (2021) for voxel representation. More details are given in Appendix B.2 and B.1.

## 4.2 COMPARATIVE STUDY

Table 1: Comparison with the state-of-the-art LiDAR SemiSL methods. All representations are evaluated with standalone architectures during inference. The **best** and underline second best result for each setting of label proportion is highlighted in **bold** and underline. $\star$: reproduced results using the released codebase. IT2$_R$ and IT2$_V$ denote models trained together with range-view and voxel representations, respectively

| Repr. | Method | nuScenes | | | | SemanticKITTI | | | | ScribbleKITTI | | | |
|---|---|---|---|---|---|---|---|---|---|---|---|---|---|
| | | 1% | 10% | 20% | 50% | 1% | 10% | 20% | 50% | 1% | 10% | 20% | 50% |
| F-Range | sup. only | 51.9 | 68.1 | 70.9 | 74.6 | 44.9 | 60.4 | 61.8 | 63.1 | 42.4 | 53.5 | 55.1 | 57.0 |
| | PolarMix Xiao et al. (2022) | 55.6 | 69.6 | 71.0 | 73.8 | 50.1 | 60.9 | 62.0 | 63.8 | 43.2 | 55.0 | 56.1 | 57.3 |
| | LaserMix Kong et al. (2023c) | 58.7 | 71.5 | 72.3 | 75.0 | 52.9 | 62.9 | 63.2 | 65.0 | 45.8 | 56.8 | 57.7 | 59.0 |
| | FrustumMix Xu et al. (2023) | 61.2 | 72.2 | 74.6 | 75.4 | 55.8 | 64.8 | 65.2 | 65.4 | 46.6 | 57.0 | 59.5 | **61.2** |
| | *CoLLiS* (Ours) | **63.2** | **74.2** | **74.8** | **75.8** | **56.0** | 64.3 | 64.9 | **66.2** | **47.6** | **59.9** | **60.1** | 60.7 |
| Polar | sup. only | *46.5 | 58.5 | 63.9 | 68.4 | *41.6 | *50.2 | *51.8 | *53.0 | *36.2 | *46.5 | *48.1 | *49.6 |
| | LaserMix Kong et al. (2023c) | *53.6 | *64.5 | *66.5 | *69.3 | - | - | - | - | - | - | - | - |
| | IT2$_R$ Liu et al. (2025) | *52.9 | 64.8 | 67.9 | 70.6 | - | - | - | - | - | - | - | - |
| | IT2$_V$ Liu et al. (2025) | *54.4 | 66.3 | 69.1 | 71.6 | - | - | - | - | - | - | - | - |
| | *CoLLiS* (Ours) | **57.9** | **68.4** | 68.6 | **70.8** | **46.8** | **53.3** | **54.0** | **55.5** | **42.0** | **51.1** | **51.4** | **51.7** |
| Voxel | sup. only | 50.9 | 65.9 | 66.6 | 71.2 | 45.4 | 56.1 | 57.8 | 58.7 | 39.2 | 48.0 | 52.1 | 53.8 |
| | CBST Zou et al. (2018) | 53.0 | 66.5 | 69.6 | 71.6 | 48.8 | 58.3 | 59.4 | 59.7 | 41.5 | 50.6 | 53.3 | 54.5 |
| | CPS Chen et al. (2021b) | 52.9 | 66.3 | 70.0 | 72.5 | 46.7 | 58.7 | 59.6 | 60.5 | 41.4 | 51.8 | 53.9 | 54.8 |
| | LaserMix Kong et al. (2023c) | 55.3 | 69.9 | 71.8 | 73.2 | 50.6 | 60.0 | 61.9 | 62.3 | 44.2 | 53.7 | 55.1 | 56.8 |
| | IT2 Liu et al. (2025) | 57.5 | 72.0 | 73.6 | 74.1 | 52.0 | 61.4 | 62.1 | 62.5 | 47.9 | 56.7 | 57.5 | 58.3 |
| | *CoLLiS* (Ours) | **61.1** | **72.9** | 73.4 | **74.5** | **53.2** | **63.1** | **63.6** | **64.0** | 47.6 | **58.6** | **58.8** | **59.0** |

**Improvements over baseline** We evaluate *CoLLiS* against existing LiDAR SemiSL approaches across diverse input representations, datasets, and label proportions (Tab. 1). Compared with single-representation methods such as LaserMix Kong et al. (2023c) and FrustumMix Xu et al. (2023), *CoLLiS* consistently achieves higher performance across most settings. The advantages are most evident in annotation-scarce scenarios, where confirmation bias is particularly severe: *CoLLiS* delivers strong gains at 1% and 10% label ratios on nuScenes Fong et al. (2022) and SemanticKITTI Behley et al. (2019), while also achieving consistent improvements on the sparsely annotated ScribbleKITTI Unal et al. (2022). These results demonstrate that collaborative learning across multiple representations effectively mitigates confirmation bias and achieves substantial advances in LiDAR SemiSL.

**Qualitative results** We qualitatively compare our approach with the dual-representation baseline (IT2) in Fig. 4. Notably, our method demonstrates superior segmentation accuracy.

## 4.3 ABLATION STUDY

**Component designs** Tab. 2 presents the ablation of *CoLLiS* on FRNet Xu et al. (2023) and Cylinder3D Zhu et al. (2021) under 1% and 20% label protocols. With 1% labels, both models gain substantially (FRNet: +4.7%, Cylinder3D: +3.3%), but benefits diminish at 20%. Incorporating two adaptive reliability factors improves robustness across two label ratios significantly, while confidence regularization Zou et al. (2019) and mixing-based augmentation add further gains (up to +3.0%). Scaling collaboration to three representations with polar images Zhang et al. (2020) yields additional improvements. Overall, adaptive parameterization, dynamic augmentation, and multi-representation collaboration are the pivotal components for performance gain.

**Dynamic augmentation** As shown in Fig. 5 (left), both CL and fixed $q_m$ are highly sensitive to hyper-parameter initialization, requiring careful tuning to avoid performance degradation. In contrast, CDA maintains stable performance across diverse initialization settings, demonstrating parameter-agnostic adaptation. Its robustness arises from dynamically adjusting mixing intensity based on inter-student consensus, which enables reliable and adaptive training.

We further examine the effect of step size in CDA (Fig. 5, right). A step size of 50 yields the best performance, indicating that frequent adjustments of augmentation intensity help prevent overfitting to a fixed perturbation level and thus improve generalization. However, excessively small step sizes

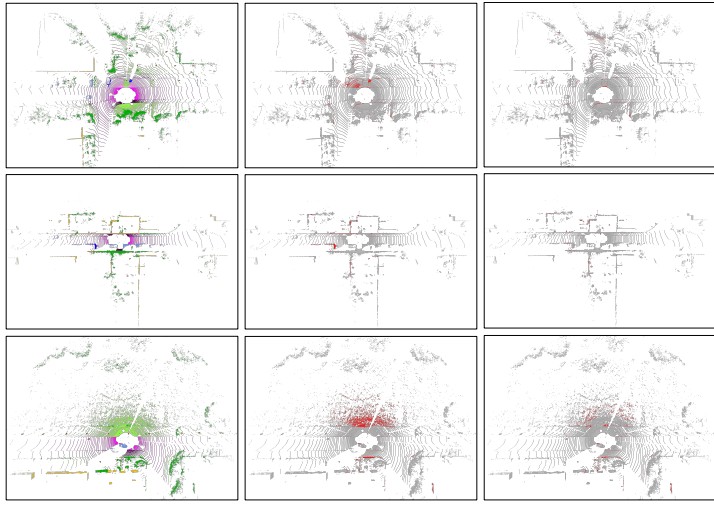

(a) Ground-truth    (b) IT2 Liu et al. (2025)    (c) *CoLLiS*

Figure 4: We qualitatively evaluate Cylinder3D Zhu et al. (2021) on SemanticKITTI. Predictions are obtained from models trained under 10% label protocol. Ground-truth labels are color-coded based on class categories. Incorrect predictions are shown in red, while correct predictions are shown in gray.

Table 2: Full ablation study on the nuScenes Fong et al. (2022) dataset. Starting from the results of supervised training, we regard the collaborative learning with naive online mutual distillation as the baseline. AR and RR denote absolute and relative reliability for pseudo-labeling. CDA representes consensus-driven augmentation.

| Co. | AR | RR | $L_{reg}$ | CDA | +Polar | FRNet | | Cylinder3D | |
|---|---|---|---|---|---|---|---|---|---|
| | | | | | | 1% | 20% | 1% | 20% |
| sup. only | | | | | | 51.9 | 70.9 | 50.9 | 66.6 |
| ✓ | | | | | | 56.6 | 71.0 | 54.2 | 66.2 |
| ✓ | ✓ | | | | | 57.5 | 71.6 | 55.5 | 67.4 |
| ✓ | ✓ | ✓ | | | | 58.8 | 72.3 | 56.5 | 69.0 |
| ✓ | ✓ | ✓ | ✓ | | | 59.5 | 72.8 | 57.3 | 70.4 |
| ✓ | ✓ | ✓ | ✓ | ✓ | | 62.5 | 74.4 | 59.2 | 73.1 |
| ✓ | ✓ | ✓ | ✓ | ✓ | ✓ | 63.2 | 74.8 | 61.1 | 73.4 |

risk biased updates due to insufficient samples. Setting the step size to 50 balances training stability with the flexibility of dynamic augmentation.

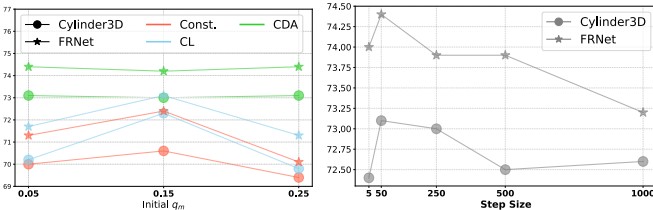

Figure 5: Ablation study on initialization of mixing probability $q_m$ (left) and step size of CDA (right) with nuScenes (20% labels).

**Post-training ensemble** During inference, all collaborative models can optionally be ensembled to refine predictions. As shown in Tab. 3, our confidence-based approach consistently outperforms naive linear combination across all scenarios. The substantial performance discrepancies among different LiDAR representations and architectures suggest that selecting the most reliable prediction often yields superior results.

Beyond its simplicity, the ensemble offers further practical benefits. Ensemble predictions can be treated as high-quality pseudo-labels for offline distillation. To verify this, we conducted an experiment where a fourth lightweight network (FIDNet Zhao et al. (2021b)) was distilled from the ensemble outputs of three collaboratively trained models. This two-stage extension is particularly useful when deploying small models for real-time applications, where standalone performance may fall short or direct collaboration with other representations is hindered by large performance gaps. As demonstrated in Tab. 4, this extension provides an effective solution without modifying the training framework, underscoring the broader utility of our method beyond direct ensemble use.

Table 3: Evaluation of **post-training ensemble** approaches. As reference, we provide the results of best performed model from *CoLLiS* evaluated without ensemble in the first row.

| Method | nuScenes | | | SemanticKITTI | | | ScribbleKITTI | | |
|---|---|---|---|---|---|---|---|---|---|
| | 1% | 20% | 50% | 1% | 20% | 50% | 1% | 20% | 50% |
| w/o ensemble | 63.2 | 74.2 | 75.8 | 56.0 | 64.3 | 66.2 | 47.6 | 59.9 | 60.7 |
| Ours | 64.5 (+1.3) | 75.5 (+1.3) | 76.2(+0.4) | 57.6 (+1.6) | 66.4 (+2.1) | 66.8 (+0.6) | 49.0(+1.4) | 60.7(+0.8) | 61.8 (+1.1) |
| Linear | 62.3 (-0.9) | 74.4 (+0.2) | 73.2 (-2.6) | 56.6 (+0.6) | 62.8 (-1.5) | 64.8 (-1.4) | 46.6 (-1.0) | 58.8 (-1.1) | 57.2 (-3.5) |

Table 4: Results of *CoLLiS* with the two-stage extension. $(+\Delta)$ indicates the improvement gained by extending *CoLLiS* with offline distillation.

| Repr. | Method | nuScenes | | SemanticKITTI | |
|---|---|---|---|---|---|
| | | 1% | 10% | 1% | 10% |
| Range | sup. only | 38.3 | 57.5 | 36.2 | 52.2 |
| | CPS Chen et al. (2021a) | 40.7 | 60.8 | 36.5 | 52.3 |
| | LaserMix Kong et al. (2023c) | 49.5 | 68.2 | 43.4 | 58.8 |
| | IT2 Liu et al. (2025) | 56.5 | 71.3 | 51.9 | 60.3 |
| | *CoLLiS* | 57.8 | 70.8 | 50.3 | 61.0 |
| | *CoLLiS* + offline distillation | 60.1 (+2.3) | 73.5 (+2.7) | 53.5 (+3.2) | 63.9 (+2.9) |

## 4.4 DISCUSSION

In this section, we analyze *CoLLiS* from several perspectives to provide deeper insights.

**Do we need collaboration?** We first examine the necessity of collaboration and assess whether our proposed setting is more effective for LiDAR semi-supervised learning. In Tab. 20, we compare *CoLLiS* with (i) ensembling individually trained models and (ii) models trained under the standard collaborative setup (DML) Zhang et al. (2018). Our method outperforms both alternatives by a substantial margin. Notably, even without using the ensemble, *CoLLiS* still achieves superior performance, demonstrating the effectiveness of the proposed collaborative design. Furthermore, we analyze performance dynamics over training progress in Fig. 11a. Within DML setup, performance collapses midway through training and continues to decline for all models, suggesting that conventional collaborative learning struggles to recover once representations begin to drift. In contrast, *CoLLiS* maintains stable and consistent improvements across all representations throughout training, highlighting its robustness and training stability.

**Mitigating confirmation bias with *CoLLiS*** Because overall performance (mIoU) does not fully capture how well our method mitigates confirmation bias, we further examine specific long-tail classes, which contain substantially fewer annotations, making them especially vulnerable to over-confidence toward non-long-tail categories. We focus on two classes: bicycle and construction vehicle from nuScenes dataset, which exhibit the weakest supervised performance at the 1% label ratio. As shown in Fig. 6, our method consistently improves segmentation performance on these long-tail classes over other state-of-the-art approaches. These improvements highlight its effectiveness in alleviating confirmation bias under low-resource settings. Moreover, through distillation across multiple LiDAR representations, our method substantially enhances the supervised performance of each collaborative model on these classes (blue-shaded area in Fig. 6(a)).

Nevertheless, the bicycle class remains particularly challenging. Since all models perform poorly even under full supervision, we hypothesize that the issue arises primarily from severe class imbalance, as the class constitutes only 0.01% of the total annotations. As a preliminary step toward addressing this issue, we explored class-rebalancing strategies such as long-tail class pasting Xiao et al. (2022). While these techniques alone yield moderate improvements, their performance remains clearly below that of our collaborative framework. However, when combined with *CoLLiS*, they offer complementary gains. This synergy highlights a promising direction for future work. A more detailed discussion is provided in Appendix D.4.

Beyond evaluating long-tail performance, we directly measure confirmation bias during training to quantitatively assess whether our method mitigates this issue. Following Arazo et al. (2020), we measure the average certainty of incorrect predictions using $-\frac{1}{N^\star} \sum_{n=1}^{N^\star} \mathbf{U}^T \log(P^\star)$, where $N^\star$ is the number of incorrect pseudo-labels, $P^\star$ is their predicted probability distribution and $\mathbf{U}$ is a uniform prior. This metric reflects how confident the model is when it is wrong. We track this metric throughout training and present the results in Fig. 11b. The results show that our method consistently reduces the certainty of incorrect predictions, whereas the certainty steadily increases when models are trained under the standard collaborative setup.

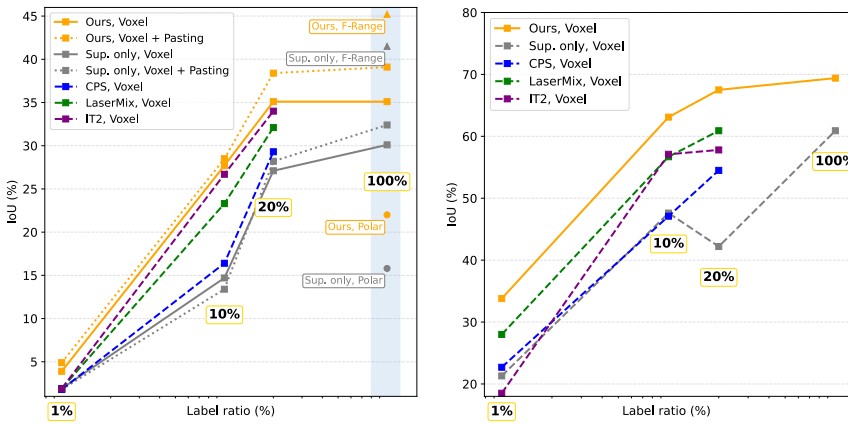

(a) Bicycle (0.01% annotations)  (b) Construction vehicle (0.13% annotations)

Figure 6: IoU of long-tail classes in the nuScenes Fong et al. (2022) dataset under varying label ratios. Pasting denotes long-tail class pasting Xiao et al. (2022).

**Pseudo-label in *CoLLiS*** To further verify that the models are truly collaborating, we examine the retention rate of pseudo-labels (Fig. 10a) and their accuracy (Fig. 10b) throughout training. These plots show that each representation retains a non-trivial portion of its pseudo-labels after thresholding, and that the overall retention rate gradually increases as training progresses, supporting our hypothesis that pseudo-label reliability improves over time. Although stronger representations retain a larger proportion of pseudo-labels, weaker ones are not entirely discarded. Because thresholding is performed independently for each student, even weaker models can preserve valid pseudo-labels in regions where they are confident. This design ensures that collaboration remains active across all views rather than collapsing toward a single dominant modality.

**Efficiency of *CoLLiS* in training** As *CoLLiS* employs three distinct backbones for collaborative learning, it is essential to evaluate its efficiency. Tab. 5 reports the training time and memory consumption. Despite leveraging multiple representations, our method requires substantially less memory than IT2 Liu et al. (2025) and achieves a training speed that is about 40% faster. Even when compared with LaserMix Kong et al. (2023c), a single-representation approach, *CoLLiS* exhibits comparable memory usage. These results highlight the efficiency advantage of our streamlined single-step design when scaling to multiple LiDAR representations.

Table 5: **Training efficiency**. The backend of LaserMix is Cylinder3D Zhu et al. (2021). The latency is measured by a single Tesla V100 GPU. Measurement of *CoLLiS* is the sum of all collaborative models.

| Method | Latency | Memory |
|---|---|---|
| LaserMix Kong et al. (2023c) | - | 10.2G |
| IT2 Liu et al. (2025) | 1080ms | 13.11G |
| *CoLLiS* | 654ms | 11.06G |

**Representation choice** Frustum-range representation is a variant of the range-view representation. Therefore, we compare these two choices in Tab. 6. Using the standard range-view representation results in a performance drop. This stems from the inherent limitations of spherical projection: multiple 3D points may collapse onto the same pixel, retaining only the nearest point and discarding others Kong et al. (2023a). Such spatial information loss degrades pseudo-label quality and propagates errors during cross-model distillation. These findings highlight the importance of preserving 3D structural fidelity when selecting LiDAR representations for collaboration, and we regard this as a limitation to address in future work.

Table 6: **Representation choice**. *CoLLiS*-triple denotes the default setup with three participants, while *CoLLiS*-dual uses only two representations. Experiments are conducted on the nuScenes Fong et al. (2022) dataset with 10% labeled data.

| Method | Frustum-Range | Range | PolarNet | Cylinder3D |
|---|---|---|---|---|
| *CoLLiS*-multi | ✓ | | 68.4 | 72.9 |
| *CoLLiS*-multi | | ✓ | 67.1 | 71.7 |
| *CoLLiS*-dual | ✓ | | - | 72.3 |
| *CoLLiS*-dual | | ✓ | - | 71.2 |

**Framework scalability** To assess how well the framework scales with additional collaborators, we introduce a fourth representation using Point Transformer V3 (PTv3) Wu et al. (2024) as the backbone and train *CoLLiS* with varying numbers of representations. Results are shown in Tab. 19. We fix the label ratio at 1%, where PTv3 also underperforms, to more rigorously assess the effectiveness of *CoLLiS* under limited supervision. As shown in Tab. X, adding more representations yields consistent gains across the board. Strong representations (e.g., Voxel, F-Range) see small but steady improvements, while weaker representations (e.g., Polar, Raw Points) experience the most significant benefits. This pattern suggests saturation in returns for already strong models but substantial advantages for underperforming ones.

**Other settings** Following prior work Li et al. (2023); Li & Dong (2024), we reproduced our method under a significant data split where frame overlap is minimized (Tab. 7). In this setting, *CoLLiS* consistently outperforms competing methods. We also evaluated *CoLLiS* under a fully supervised setting (Tab. 19). Even with full label availability, all three collaborative models achieve clear improvements over their standalone baselines.

Table 7: Results on significant data split. All models are evaluated independently.

| Method | nuScenes | | SemanticKITTI | |
|---|---|---|---|---|
| | 1% | 10% | 1% | 10% |
| GPC Jiang et al. (2021) | - | - | 54.1 | 62.0 |
| Lim3D Li et al. (2023) | - | - | 58.4 | 62.2 |
| DDSemi Li & Dong (2024) | 58.1 | 70.2 | 59.3 | 65.1 |
| *CoLLiS* (Voxel) | 62.3 | 74.1 | 58.8 | 65.5 |
| *CoLLiS* (F-Range) | **64.6** | **74.9** | **60.5** | **66.1** |

Table 8: **Supervised training** results on nuScenes dataset. We do not apply test time augmentation or ensembling during inference.

| Method | FRNet | PolarNet | Cylinder3D |
|---|---|---|---|
| sup. only | 77.7 | 70.4 | 72.1 |
| *CoLLiS* | 78.4 (+0.7) | 73.2 (+2.8) | 75.3 (+3.2) |

**Future works** While effective, *CoLLiS* still faces challenges with extremely rare long-tail classes due to performance degradation shared across all collaborative models. Future work will explore incorporating rebalancing technique to enhance the generalization ability Chang et al. (2024); Wei et al. (2021). Another promising direction is the use of contrastive learning across multiple representations to enhance cross-view consistency Liu et al. (2025). However, the substantial computational cost associated with contrastive objectives may hinder the scalability of multi-representation training. Moreover, *CoLLiS* does not include multi-modality or temporal cues for training. Integrating collaborative learning with richer supervision sources, e.g. image-LiDAR alignment Sautier et al. (2022); Liu et al. (2023), language guidance Kong et al. (2025) and temporal consistency Lin et al. (2025); Xu et al. (2024) may further enhance label-efficient training in real-world autonomous driving applications. We leave the integration of these complementary signals into our framework as valuable directions for future exploration.

## 5 CONCLUSION

In this work, we presented *CoLLiS*, a collaborative learning framework for semi-supervised LiDAR semantic segmentation. *CoLLiS* trains multiple networks on different LiDAR representations as co-equal students within a single step. The framework not only scales multi-representation learning efficiently but also addresses confirmation bias through balanced knowledge transfer and improves generalization via adaptive data augmentation. We further show that ensembling student outputs provides additional gains. Extensive experiments on three public benchmarks show consistent improvements over state-of-the-art methods.

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

## A    THE USE OF LARGE LANGUAGE MODELS (LLMS)

We used a large language model (OpenAI's ChatGPT) solely as a writing assistant to polish the text, check grammar, and provide minor stylistic suggestions (e.g., color choices and fontsize in Fig. 6, Fig. 4 and Fig. 8) for better readability. The model was not involved in research ideation, methodology design, or experimental analysis. All technical content and experimental results are entirely the work of the authors.

## B    ADDITIONAL DETAILS

### B.1    DATASET

nuScenes Fong et al. (2022) contains 1,000 driving scenes captured by a 32-beam LiDAR, annotated with 16 semantic classes after merging similar and infrequent classes. SemanticKITTI Behley et al. (2019) consists of 22 sequences captured with a 64-beam LiDAR and includes 19 semantic classes, with sequences 00–07/09–10 (19,130 scans) for training and 08 (4,071 scans) for validation. ScribbleKITTI Unal et al. (2022) shares the same point cloud data of SemanticKITTI but replaces full annotations with sparse scribbles (8.06% labeled points for training).

### B.2    IMPLEMENTATION

The initial confidence threshold ($\delta_0$) is set to 0.95 for nuScenes Fong et al. (2022) and 0.9 for other two datasets. The initial weights of unlabeled loss ($\lambda_0$) are predefined based on scarcity level for nuScenes and SemanticKITTI Behley et al. (2019): (1, 0.5, 0.5, 0.3) for labeled proportions of (1%, 10%, 20%, 50%), while for ScribbleKITTI Unal et al. (2022), the weight is fixed at 1 due to its inherently sparse annotations.

In the *multi-representation* setting, We deploy FRNet Xu et al. (2023), Cylinder3D Zhu et al. (2021) and PolarNet Zhang et al. (2020). For FRNet, the 2D representation shape is set to $64 \times 512$ for SemanticKITTI Behley et al. (2019) and ScribbleKITTI Unal et al. (2022), and $32 \times 480$ for nuScenes Fong et al. (2022). For PolarNet and Cylinder3D, the grid size is reduced to $240 \times 180 \times 20$, following the settings in prior works Kong et al. (2023c); Liu et al. (2025) for fair comparison.

In the *dual-representation* setting, we follow the same configuration of IT2 Liu et al. (2025), where a range-view network FIDNet Zhao et al. (2021b) and Cylinder3D Zhu et al. (2021) are incorporated in the framework.

### B.3    TRAINING DETAILS

We use a batch size of 14 for both labeled and unlabled data (effective batch size is then 28 after mixing) for nuScenes Fong et al. (2022) dataset. The learning rate is set to $6e^{-3}$, and the maximum number of epochs is set to 100. For SemanticKITTI Behley et al. (2019) and ScribbleKITTI Unal et al. (2022), the batch size is reduced to 8 and learning rate is $8e^{-3}$. The maximum number of epochs is 95. For all experiments in the work, we use the AdamW Loshchilov & Hutter (2017) optimizer with a weight decay of 0.0001 and a OneCycleLR scheduler Smith & Topin (2019), and employ a single NVIDIA H200 GPU for training.

### B.4    EVALUATION METRICS

The performance is measured with average Intersection-over-Union (mIoU).

### B.5    MIXING STRATEGIES

To enhance LiDAR point cloud augmentation, we integrate three geometrically complementary mixing strategies, each tailored to exploit distinct spatial properties of LiDAR data. In Fig. 7, we visualize the point clouds mixed by different strategies. Notably, the visualization shows that mixed point clouds have distinct difference in geometry. Below, we provide technical details of every mixing strategy:

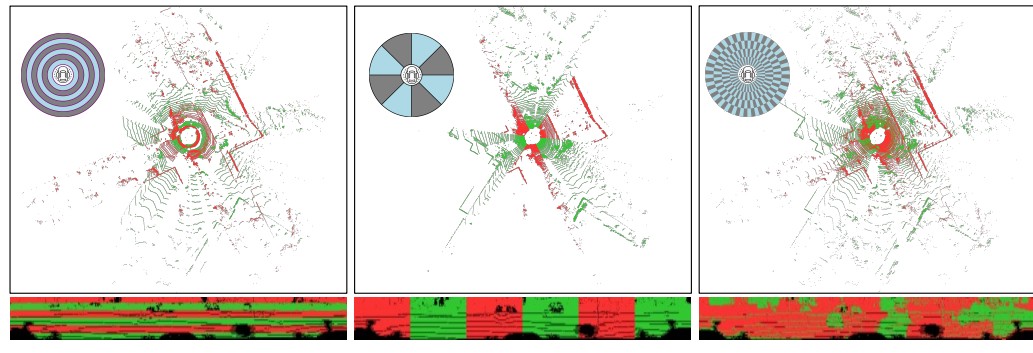

|                (a) LaserMix                |                (b) PolarMix                |        (c) Sub-cloud Shuffling        |

Figure 7: Examples of different mixing strategies using two LiDAR point clouds (distinguished by green and red). Mixed point clouds are visualized in bird's-eye view (top) and range view (bottom).

1. LaserMix Kong et al. (2023c) partitions two scenes along elevation angle (vertical sweep axis) and interleaves their sectors while keeping the ring-like geometry of the scene. The mixing generally follows the inherent scan pattern of LiDAR sensor.

2. PolarMix Xiao et al. (2022) operates in the polar coordinate space, splitting point clouds radially and horizontally into different several regions. Mixed scenes retain intact local object geometry by constraining swaps to entire polar regions.

3. Sub-cloud Shuffling Yang & Condurache (2025) downsamples two point clouds into two sub-clouds, respectively, then randomly interleaves them. this preserves local coherence and scene integrity while fusing semantic contexts from both scenes

Overall, these three mixing strategies augment point clouds from different geometric perspectives. By combining them, we harness complementary spatial transformations that capture both local and global structures, enhancing the diversity of geometric patterns. This synergy increases the generalization capability of data augmentation within our framework, leading to more robust and adaptable semi-supervised LiDAR semantic segmentation.

## C  ADDITIONAL EXPERIMENTS

### C.1  HETEROGENEOUS *CoLLiS*

Besides development in using different representations for LiDAR semantic segmentation. parallel advancements have occurred in architectural design. Driven by the success of Transformers Vaswani (2017) in vision tasks Dosovitskiy et al. (2021); Zhao et al. (2021a), recent works increasingly replace conventional CNNs with Transformer-based backbones for LiDAR segmentation Ando et al. (2023); Kong et al. (2023a); Lai et al. (2023); Wu et al. (2024), leveraging their ability to model long-range dependencies in sparse 3D data.

To additionally evaluate the impact of architectural diversity independently of representational differences, we further test *CoLLiS* in a setting where the same LiDAR representation is processed by two networks with heterogeneous architectures. In this scenario, we integrate two range-view methods, FIDNet Zhao et al. (2021b) and RangeViT Ando et al. (2023). The shape of range-view image is set to $32 \times 1920$ for nuScenes and $64 \times 2048$ for ScribbleKITTI.

Tab. 9 demonstrates that *CoLLiS* achieves significant improvements in semi-supervised scenarios even with a single representation by integrating heterogeneous architectures (e.g., CNNs and ViTs). For instance, with 1% labeled data on ScribbleKITTI Unal et al. (2022), combined networks outperforms LaserMix Kong et al. (2023c) by +2.5% mIoU while trained on the same range-view inputs. In other settings, collaboration of heterogeneous networks exhibits the competitive performance as well. This underscores that architectural diversity drives robust representation learning in label-scarce regimes.

Table 9: Evaluation of **FIDNet** Zhao et al. (2021b) with **heterogeneous** *CoLLiS*. The performance is compared with other single-representation approaches. *: We re-implement the IT2 Liu et al. (2025) framework with heterogeneous architectures for fair comparison. The best results are highlighted in **bold**.

| Method | nuScenes Fong et al. (2022) | | | ScribbleKITTI Unal et al. (2022) | | |
|---|---|---|---|---|---|---|
| | 1% | 10% | 20% | 1% | 10% | 20% |
| sup. only | 38.3 | 57.5 | 62.7 | 33.1 | 47.7 | 49.9 |
| CPS Chen et al. (2021b) | 40.7 | 60.8 | 64.9 | 33.7 | 50.0 | 52.8 |
| LaserMix Kong et al. (2023c) | 49.5 | 68.2 | **70.6** | 38.3 | 54.4 | 55.6 |
| *IT2-Hete Liu et al. (2025) | 50.1 | **68.3** | 69.9 | 39.5 | 53.4 | **57.8** |
| *CoLLiS-Hete* | **50.3** | 67.7 | 69.6 | **40.8** | **55.1** | 56.2 |

## C.2  DYNAMIC AUGMENTATION

Tab. 10 compares adaptation strategies for mixing probability. Curriculum Learning (CL) outperforms fixed $q_m$ under moderate label scarcity (20%) but loses effectiveness under extreme scarcity (1%). In contrast, our consensus-driven augmentation (CDA) consistently achieves the best results across both label regimes and networks. For mixing strategies, combining LaserMix (*LM*), PolarMix (*PM*), and Sub-cloud Shuffling (*SS*) yields the strongest performance, as each exploits complementary geometric cues: vertical beams, radial sectors, and global scene integrity.

Table 10: Ablation study on dynamic augmentation with nuScenes Fong et al. (2022) dataset. Const. denotes that a constant value is assigned to $q_m$ (0.25 for 1% and 0.15 for 20%). For curriculum learning (CL), $(q_{m,min}, q_{m,max})$ is set to (0.2, 0.3) for 1% and (0.15, 0.25) for 20%, respectively. The step size ($N$) for CDA is fixed at 50 for both settings. {*LM, PM, SS*} are three different mixing strategies.

| **Const.** | **CL** | **CDA** | *LM* | *PM* | *SS* | FRNet | | Cylinder3D | |
|---|---|---|---|---|---|---|---|---|---|
| | | | | | | 1% | 20% | 1% | 20% |
| ✓ | | | ✓ | ✓ | ✓ | 61.1 | 72.4 | 58.0 | 70.6 |
| | ✓ | | ✓ | ✓ | ✓ | 60.9 | 73.1 | 58.2 | 72.3 |
| | | ✓ | ✓ | ✓ | ✓ | 62.5 | 74.4 | 59.2 | 73.1 |
| | | | ✓ | ✓ | | 60.4 | 73.2 | 58.2 | 72.1 |
| | | | ✓ | ✓ | ✓ | 61.8 | 74.4 | 59.0 | 72.9 |

## C.3  WEIGHT OF UNLABELED LOSS

We further examine the effect of initial unlabeled loss weights in Tab. 11. An improper choice consistently degrades segmentation performance across all student models. Setting the weight too high causes overfitting to incorrect pseudo-labels early in training, while setting it too low prevents sufficient knowledge transfer, leading to overfitting on the limited labeled data. This underscores the importance of careful tuning to balance knowledge transfer and stability in *CoLLiS*. We also evaluate adaptive distillation weights in Tab. 12. The results show clear performance gains, confirming that softly resolving label conflicts is crucial for robust collaborative training.

Table 11: Ablation study on the initial weight of unlabeled loss ($\lambda_0$). Results are reported on nuScenes Fong et al. (2022) dataset with 10% labels.

| $\lambda_0$ | FRNet Xu et al. (2023) | PolarNet Zhang et al. (2020) | Cylinder3D Zhu et al. (2021) |
|---|---|---|---|
| 0.2 | 73.5 | 66.9 | 72.1 |
| 0.5 | 74.2 | 68.4 | 72.9 |
| 0.8 | 73.1 | 67.4 | 72.3 |

Additionally, we investigate the impact of adaptive distillation weights in Tab. 12. Under our collaborative setup, we intentionally avoid enforcing strict consensus on pseudo-labels to prevent models from collapsing toward the strongest model, thereby mitigating confirmation bias. However, this design can introduce noise when disagreements arise among students. To address this, we introduce adaptive weights $\omega$ to softly resolve such conflicts. As shown in Tab. 12, these adaptive weights play a critical role in enhancing collaborative synergy and reducing the effect of conflict-induced noise.

Table 12: Ablation study on the adaptive distillation weights ($\omega_{s_i \to s_j}$). We evaluated Cylinder3D Zhu et al. (2021) on nuScenes Fong et al. (2022) dataset with 1% and 10% labels.

| $\omega_{s_i \to s_j}$ | 1% | 10% |
|---|---|---|
| Fixed = 1 | 59.0 | 71.2 |
| Adaptive | 61.1 | 72.9 |

Table 13: The class-wise IoU results in nuScenes Fong et al. (2022) dataset among different partition protocol. The mIoU results are highlighted in red. 100% label proportion denotes the fully supervised training results.

| Repr. | prop. | mIoU | barr | bicy | bus | car | const | moto | ped | cone | trail | truck | driv | othe | walk | terr | manm | veg |
|---|---|---|---|---|---|---|---|---|---|---|---|---|---|---|---|---|---|---|
| F-Range | 1% | **63.2** | 66.5 | 3.1 | 78.4 | 86.9 | 14.6 | 66.3 | 72.3 | 50.2 | 35.8 | 60.0 | 95.3 | 65.1 | 69.5 | 73.9 | 86.2 | 86.2 |
| | 10% | **74.2** | 76.9 | 0.8 | 92.6 | 85.7 | 54.2 | 84.7 | 73.1 | 66.2 | 68.7 | 84.0 | 96.7 | 75.9 | 75.9 | 77.0 | 88.1 | 86.7 |
| | 20% | **74.8** | 77.9 | 0.7 | 93.4 | 91.7 | 55.2 | 85.4 | 76.2 | 67.1 | 67.6 | 83.6 | 96.7 | 74.5 | 75.5 | 76.6 | 88.3 | 86.6 |
| | 50% | **75.8** | 77.6 | 35.1 | 92.7 | 91.1 | 50.4 | 82.5 | 73.3 | 66.1 | 65.2 | 82.2 | 96.5 | 73.1 | 74.5 | 76.3 | 88.8 | 87.4 |
| | 100% | **78.4** | 78.3 | 45.2 | 92.6 | 91.7 | 58.1 | 84.4 | 77.9 | 67.8 | 71.1 | 83.5 | 96.7 | 77.3 | 76.1 | 77.1 | 89.0 | 87.5 |
| Polar | 1% | **57.9** | 59.5 | 2.1 | 78.4 | 83.8 | 7.8 | 61.4 | 51.7 | 41.0 | 31.7 | 58.8 | 93.1 | 60.3 | 63.8 | 69.2 | 82.7 | 81.8 |
| | 10% | **68.4** | 71.4 | 11.3 | 90.0 | 87.6 | 39.1 | 69.7 | 58.5 | 51.3 | 63.6 | 77.9 | 94.9 | 69.1 | 70.4 | 71.8 | 85.2 | 82.9 |
| | 20% | **68.6** | 72.8 | 1.6 | 90.9 | 88.2 | 43.0 | 73.1 | 57.8 | 50.4 | 64.6 | 78.6 | 94.9 | 70.6 | 70.9 | 72.1 | 85.4 | 82.8 |
| | 50% | **70.8** | 72.3 | 15.9 | 92.1 | 88.4 | 47.5 | 75.4 | 58.4 | 52.4 | 67.6 | 79.6 | 95.0 | 73.1 | 71.7 | 72.3 | 85.3 | 85.2 |
| | 100% | **73.2** | 75.4 | 22.0 | 93.3 | 89.7 | 51.5 | 76.3 | 60.2 | 62.2 | 68.3 | 79.2 | 95.7 | 73.8 | 73.5 | 75.4 | 88.4 | 86.7 |
| Voxel | 1% | **61.1** | 64.2 | 3.9 | 77.2 | 85.4 | 18.8 | 63.0 | 63.7 | 47.4 | 33.8 | 58.5 | 94.0 | 61.3 | 65.2 | 71.0 | 85.3 | 85.0 |
| | 10% | **72.9** | 73.9 | 27.7 | 91.4 | 89.9 | 46.4 | 78.6 | 68.3 | 60.3 | 63.1 | 80.4 | 95.4 | 69.8 | 71.4 | 73.6 | 87.9 | 86.3 |
| | 20% | **73.4** | 74.5 | 35.4 | 92.7 | 90.6 | 45.8 | 80.5 | 69.0 | 60.2 | 67.5 | 81.6 | 95.6 | 72.1 | 72.9 | 75.0 | 87.9 | 86.1 |
| | 50% | **74.5** | 75.8 | 30.1 | 93.0 | 90.8 | 51.0 | 80.1 | 71.7 | 62.1 | 69.7 | 83.4 | 95.8 | 74.7 | 73.8 | 75.3 | 87.9 | 86.3 |
| | 100% | **75.3** | 75.7 | 35.1 | 92.7 | 90.7 | 49.7 | 81.5 | 70.7 | 62.2 | 69.4 | 83.0 | 95.6 | 74.4 | 73.0 | 75.1 | 88.4 | 86.7 |

Table 14: The class-wise IoU results in SemanticKITTI Behley et al. (2019) dataset on different label proportions. The mIoU results are highlighted in red.

| Repr. | prop. | mIoU | car | bicy | moto | truck | o.veh | ped | b.cyc | m.cyc | road | park | walk | o.gro | build | fence | veg | trunk | terr | pole | sign |
|---|---|---|---|---|---|---|---|---|---|---|---|---|---|---|---|---|---|---|---|---|---|
| F-Range | 1% | **56.0** | 91.7 | 16.8 | 48.0 | 66.0 | 50.7 | 61.8 | 68.2 | 0.0 | 90.7 | 50.9 | 73.6 | 0.9 | 85.2 | 42.7 | 86.9 | 44.9 | 73.4 | 60.2 | 41.9 |
| | 10% | **64.3** | 95.7 | 19.1 | 70.1 | 87.3 | 58.8 | 73.1 | 84.8 | 0.0 | 96.0 | 60.5 | 84.0 | 2.3 | 89.6 | 66.2 | 86.5 | 65.8 | 71.9 | 62.4 | 47.4 |
| | 20% | **64.9** | 95.5 | 36.3 | 59.5 | 91.7 | 58.9 | 71.4 | 84.0 | 0.0 | 95.8 | 58.7 | 84.2 | 6.7 | 90.9 | 68.8 | 86.5 | 65.5 | 72.0 | 63.3 | 42.7 |
| | 50% | **66.2** | 95.2 | 47.9 | 71.8 | 92.0 | 51.4 | 73.5 | 81.2 | 0.0 | 96.0 | 61.1 | 83.9 | 12.9 | 89.9 | 63.9 | 87.3 | 65.8 | 73.5 | 63.0 | 47.5 |
| Polar | 1% | **46.8** | 91.2 | 15.5 | 21.1 | 24.6 | 0.3 | 36.0 | 62.4 | 0.0 | 90.8 | 35.7 | 75.5 | 1.1 | 80.3 | 53.2 | 79.8 | 56.7 | 70.8 | 62.1 | 32.2 |
| | 10% | **53.3** | 90.5 | 20.7 | 41.8 | 78.8 | 28.2 | 31.1 | 77.1 | 2.2 | 90.9 | 51.2 | 75.1 | 0.3 | 87.0 | 40.3 | 83.3 | 53.5 | 67.9 | 55.8 | 36.7 |
| | 20% | **54.0** | 91.6 | 23.7 | 45.7 | 75.3 | 29.9 | 35.7 | 75.3 | 0.0 | 91.0 | 52.9 | 74.9 | 0.2 | 88.1 | 43.5 | 84.0 | 52.4 | 70.7 | 54.4 | 36.3 |
| | 50% | **55.5** | 94.1 | 25.7 | 47.3 | 77.0 | 28.8 | 39.4 | 77.0 | 0.0 | 90.7 | 52.2 | 74.5 | 0.5 | 87.5 | 51.9 | 85.7 | 51.4 | 67.1 | 65.2 | 37.9 |
| Voxel | 1% | **53.2** | 91.2 | 21.9 | 43.4 | 66.1 | 29.5 | 32.6 | 74.1 | 0.6 | 90.9 | 49.9 | 74.8 | 0.0 | 87.7 | 47.6 | 84.2 | 51.3 | 70.8 | 53.4 | 40.8 |
| | 10% | **63.1** | 94.0 | 38.2 | 59.3 | 82.4 | 52.0 | 72.2 | 81.0 | 0.9 | 92.5 | 51.8 | 78.4 | 7.8 | 91.1 | 61.8 | 86.6 | 67.8 | 71.4 | 63.4 | 45.8 |
| | 20% | **63.6** | 93.4 | 46.1 | 68.0 | 87.8 | 35.7 | 72.8 | 87.0 | 9.2 | 92.6 | 45.8 | 78.2 | 3.3 | 90.4 | 58.7 | 87.1 | 65.5 | 72.8 | 63.6 | 49.6 |
| | 50% | **64.0** | 94.2 | 50.3 | 67.7 | 89.8 | 43.2 | 71.2 | 88.1 | 4.2 | 92.9 | 49.0 | 78.5 | 0.4 | 90.6 | 60.0 | 86.5 | 67.1 | 71.0 | 63.6 | 48.1 |

## C.4 OUT-OF-DISTRIBUTIONS GENERALIZATION

To assess the effectiveness of our method under distribution shift, we collaboratively train models using full annotations and clean data, and then directly evaluate each individual model on Robo3D Kong et al. (2023b), which contains data simulated under various corruption scenarios. As shown in Tab. 18, *CoLLiS* outperforms the baseline on two of the three backbones. For FRNet, the strongest backbone among the three, *CoLLiS* achieves performance that remains comparable to its baseline counterpart.

# D ADDITIONAL RESULTS

## D.1 DETAILED RESULTS

We provide the detailed results of class-wise IoU in Tab. 13, Tab. 14 and Tab. 15.

## D.2 QUALITATIVE RESULTS

We provide additional qualitative results in Fig. 8.

Table 15: The class-wise IoU results in ScribbleKITTI Unal et al. (2022) dataset among different partition protocol. The mIoU results are highlighted in red.

| Repr. | prop. | mIoU | car | bicy | moto | truck | o.veh | ped | b.cyc | m.cyc | road | park | walk | o.gro | build | fence | veg | trunk | terr | pole | sign |
|---|---|---|---|---|---|---|---|---|---|---|---|---|---|---|---|---|---|---|---|---|---|
| F-Range | 1% | 47.6 | 90.3 | 30.8 | 27.4 | 27.5 | 4.4 | 37.1 | 66.3 | 0.0 | 83.7 | 36.3 | 71.9 | 3.3 | 87.0 | 39.9 | 80.3 | 58.3 | 66.1 | 56.1 | 38.2 |
| | 10% | 59.9 | 94.4 | 28.6 | 61.9 | 83.6 | 42.7 | 64.4 | 80.4 | 0.0 | 87.1 | 46.7 | 75.0 | 0.6 | 89.8 | 54.4 | 84.7 | 65.7 | 68.9 | 63.5 | 46.8 |
| | 20% | 60.1 | 94.6 | 30.6 | 60.9 | 92.2 | 46.4 | 64.2 | 83.2 | 0.0 | 86.3 | 46.1 | 74.1 | 0.5 | 88.7 | 49.9 | 83.5 | 64.4 | 65.9 | 63.4 | 47.9 |
| | 50% | 60.7 | 95.1 | 31.1 | 58.6 | 88.4 | 50.6 | 63.3 | 85.4 | 0.0 | 86.6 | 43.1 | 74.3 | 0.7 | 89.6 | 52.3 | 84.9 | 66.2 | 69.7 | 63.8 | 48.7 |
| Polar | 1% | 42.0 | 89.2 | 23.2 | 18.0 | 18.9 | 3.1 | 20.9 | 61.6 | 0.3 | 82.0 | 23.5 | 66.8 | 0.7 | 86.9 | 27.4 | 77.9 | 49.5 | 61.6 | 52.4 | 33.4 |
| | 10% | 51.1 | 91.1 | 16.1 | 49.1 | 52.6 | 34.0 | 34.2 | 81.1 | 0.0 | 84.7 | 40.9 | 69.4 | 0.1 | 87.4 | 33.9 | 81.8 | 55.6 | 68.1 | 55.5 | 36.3 |
| | 20% | 51.4 | 91.1 | 15.5 | 48.7 | 57.8 | 34.2 | 33.5 | 81.0 | 0.0 | 84.7 | 40.8 | 69.4 | 0.1 | 87.5 | 33.9 | 81.7 | 55.4 | 68.0 | 55.4 | 35.8 |
| | 50% | 51.7 | 91.3 | 18.2 | 47.4 | 72.5 | 32.0 | 36.3 | 73.8 | 0.4 | 83.7 | 42.4 | 68.5 | 0.1 | 86.4 | 36.4 | 81.7 | 54.3 | 65.8 | 55.4 | 35.1 |
| Voxel | 1% | 47.6 | 90.4 | 30.8 | 12.2 | 20.9 | 2.2 | 41.1 | 71.1 | 0.2 | 82.2 | 25.4 | 68.4 | 0.7 | 88.2 | 31.9 | 80.6 | 63.2 | 65.3 | 61.5 | 39.7 |
| | 10% | 58.6 | 93.3 | 29.6 | 56.1 | 68.3 | 40.2 | 61.6 | 83.5 | 8.9 | 85.9 | 39.8 | 72.4 | 1.1 | 89.4 | 46.2 | 86.0 | 66.6 | 71.9 | 64.4 | 49.2 |
| | 20% | 58.8 | 93.3 | 31.3 | 59.8 | 79.4 | 38.7 | 64.4 | 82.3 | 0.0 | 85.0 | 35.4 | 72.1 | 3.6 | 89.2 | 43.8 | 85.3 | 68.1 | 71.0 | 64.0 | 49.5 |
| | 50% | 59.0 | 93.5 | 29.9 | 60.3 | 84.4 | 41.3 | 64.5 | 84.7 | 6.3 | 84.3 | 36.0 | 71.2 | 0.7 | 89.1 | 42.2 | 84.4 | 67.3 | 68.7 | 63.0 | 49.7 |

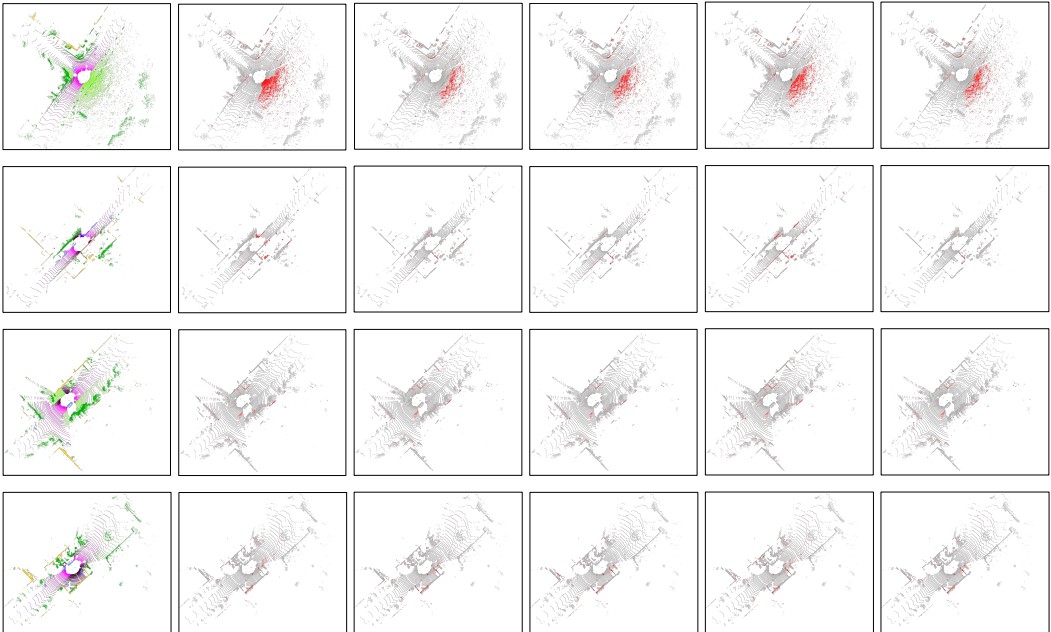

(a) Ground-truth    (b) Voxel (IT2)    (c) Voxel(Ours)    (d) F-Range(Ours)    (e) Polar(Ours)    (f) Fusion(Ours)

Figure 8: **Qualitative results** on SemanticKITTI Behley et al. (2019). All models are trained under the 10% label protocol. We use Hard Confidence Voting (HCV) to fuse students' outputs. Ground-truth labels are color-coded based on class categories. Incorrect predictions are shown in red, while correct predictions are shown in gray.

## D.3    INFERENCE COST

We report the inference cost for each model used in collaborative training in Tab. 16. In the standard setup, each model is evaluated independently, and the resulting cost reflects its respective architecture. When the optional ensemble mode is enabled during inference, the overall cost increases due to the additional forward passes of all individual models. While optional, the ensemble mode limits the deployability of our framework in real-time applications. To address this limitation, we introduce a two-stage extension in Sec. 4.4. Specifically, we treat the ensemble predictions from the three collaboratively trained models as high-quality pseudo-labels and distill them into a lightweight model. This distilled model achieves performance close to the ensemble while operating independently during inference, thereby significantly reducing computational overhead and enabling real-time deployment.

## D.4    LIMITED GAIN ON EXTREMELY LONG-TAIL CLASSES

In this section, we elaborate on why *CoLLiS* yields only marginal improvements on extremely long-tail classes. As discussed in Sec. 4.4, our method continues to struggle because all collaborating

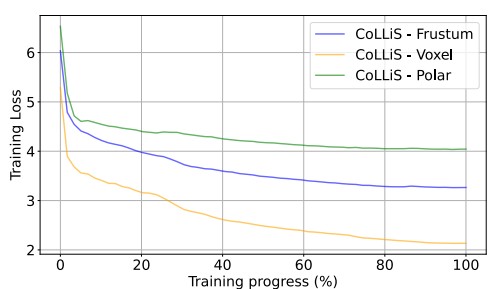

Figure 9: Training Loss Curve of *CoLLiS*. Training is conducted on nuScenes Fong et al. (2022) with 1% labeled data.

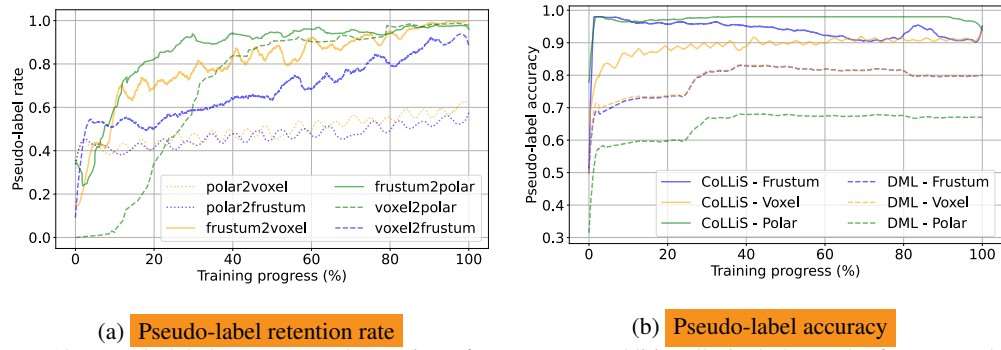

(a) Pseudo-label retention rate          (b) Pseudo-label accuracy

Figure 10: Pseudo-label accuracy and retention of *CoLLiS*. We additionally include results for DML Zhang et al. (2018). Metrics are averaged over every 500 iterations, and training is conducted on nuScenes Fong et al. (2022) with 1% labeled data.

models perform poorly even under full supervision. We quantify this limitation in Tab. 17, where all participating representations exhibit very low IoU scores due to the severe class imbalance under 1% label regime. This indicates that the bottleneck stems from intrinsic data scarcity rather than from the collaborative paradigm itself. Consequently, it is challenging for CoLLiS to close the performance gap without incorporating additional class-rebalancing strategies. Nevertheless, CoLLiS still provides consistent gains over individual representations on the rarest classes. We further illustrate these failure cases in Fig. 12. For such extremely rare classes, annotations can be highly sparse, sometimes consisting of only one or two labeled points per scene, which makes it inherently difficult for models to predict these isolated points accurately.

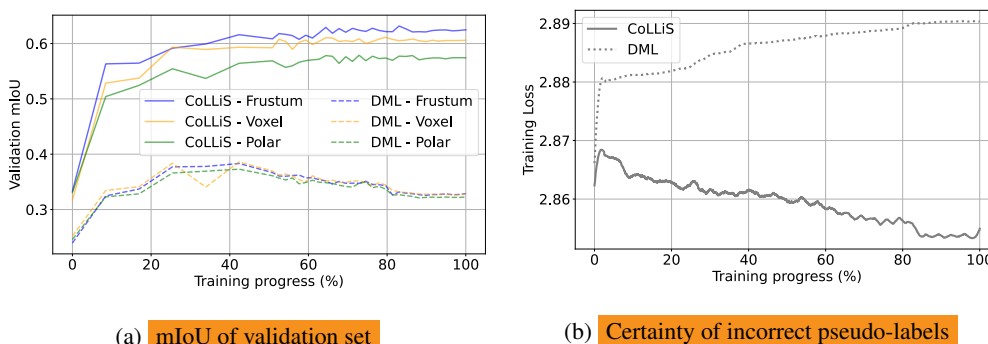

(a) mIoU of validation set      (b) Certainty of incorrect pseudo-labels

Figure 11: Performance analysis of *CoLLiS*. We additionally include results for DML Zhang et al. (2018). Metrics are averaged over every 500 iterations, and training is conducted on nuScenes Fong et al. (2022) with 1% labeled data.

Table 16: **Inference Cost** : Latency is reported as the average over 100 evaluation iterations. The optional post-training ensemble introduces an inference overhead approximately equal to the sum of the latencies of all collaborative models.

| Model | Latency | Model Size |
|---|---|---|
| Cylinder3D Zhu et al. (2021) | 102.3 ms | 12.5M |
| FRNet Xu et al. (2023) | 28.4 ms | 10.0M |
| PolarNet Xu et al. (2023) | 43.5 ms | 13.6M |
| Ensemble | 162.3 ms | - |

Table 17: We report supervised performance on the nuScenes bicycle class using 1% labeled data, where fewer than 0.01% of the annotations correspond to bicycles.

| IoU (bicycle) | sup. only | *CoLLiS* |
|---|---|---|
| Cylinder3D Zhu et al. (2021) | 1.9 | 3.9 |
| FRNet Xu et al. (2023) | 1.0 | 3.1 |
| PolarNet Xu et al. (2023) | 0.9 | 2.1 |

Table 18: Results on nuScenes-C (Robo3D Kong et al. (2023b)) . In addition to mCE, mRR, and mIoU, we also report the IoU for each corruption type. [†] indicates models trained with *CoLLiS*.

| Model | mCE↓ | mRR↑ | mIoU↑ | Fog | Weg | Snow | Motion | Beam | Cross | Echo | Sensor |
|---|---|---|---|---|---|---|---|---|---|---|---|
| FRNet | 98.6 | 77.5 | 77.7 | 69.1 | 76.6 | 69.5 | 54.5 | 68.3 | 41.4 | 58.7 | 43.1 |
| FRNet[†] | 99.8 | 77.0 | 77.1 | 68.2 | 76.9 | 69.0 | 51.8 | 66.9 | 43.6 | 59.2 | 42.5 |
| Cylinder3D | 111.8 | 72.9 | 76.2 | 59.9 | 72.7 | 58.1 | 42.1 | 64.5 | 44.4 | 60.5 | 42.2 |
| Cylinder3D[†] | 110.1 | 74.1 | 76.8 | 61.2 | 70.5 | 63.2 | 43.7 | 65.2 | 42.3 | 61.0 | 41.0 |
| PolarNet | 115.1 | 76.3 | 71.4 | 58.2 | 69.9 | 64.8 | 44.6 | 61.9 | 40.8 | 53.6 | 42.0 |
| PolarNet[†] | 111.5 | 77.9 | 74.3 | 60.4 | 70.2 | 64.2 | 47.8 | 64.3 | 42.1 | 58.4 | 43.9 |

Table 19: **Framework scalability** . We evaluate the scalability of *CoLLiS* by training it with varying numbers of representations, and we report the performance of each model independently without using the ensemble.

| #Repr. | FRNet (F-Range) | PolarNet (Polar) | Cylinder3D (Voxel) | PTv3 (Raw points) |
|---|---|---|---|---|
| **nuScenes 1%** | | | | |
| sup. only | 51.9 | 46.5 | 50.9 | 48.0 |
| 2 | 62.5 | - | 59.2 | - |
| 3 | 63.2 | 57.9 | 61.1 | - |
| 3 | **63.5** | - | 61.3 | 60.7 |
| 4 | 63.4 | **58.6** | **61.6** | **61.2** |
| **SemanticKITTI 1%** | | | | |
| sup. only | 44.9 | 41.6 | 45.4 | 41.6 |
| 2 | 55.1 | - | 51.8 | - |
| 3 | 56.0 | 46.8 | 53.2 | - |
| 3 | 56.8 | - | 54.3 | 54.8 |
| 4 | **56.8** | **48.9** | **54.7** | **55.2** |

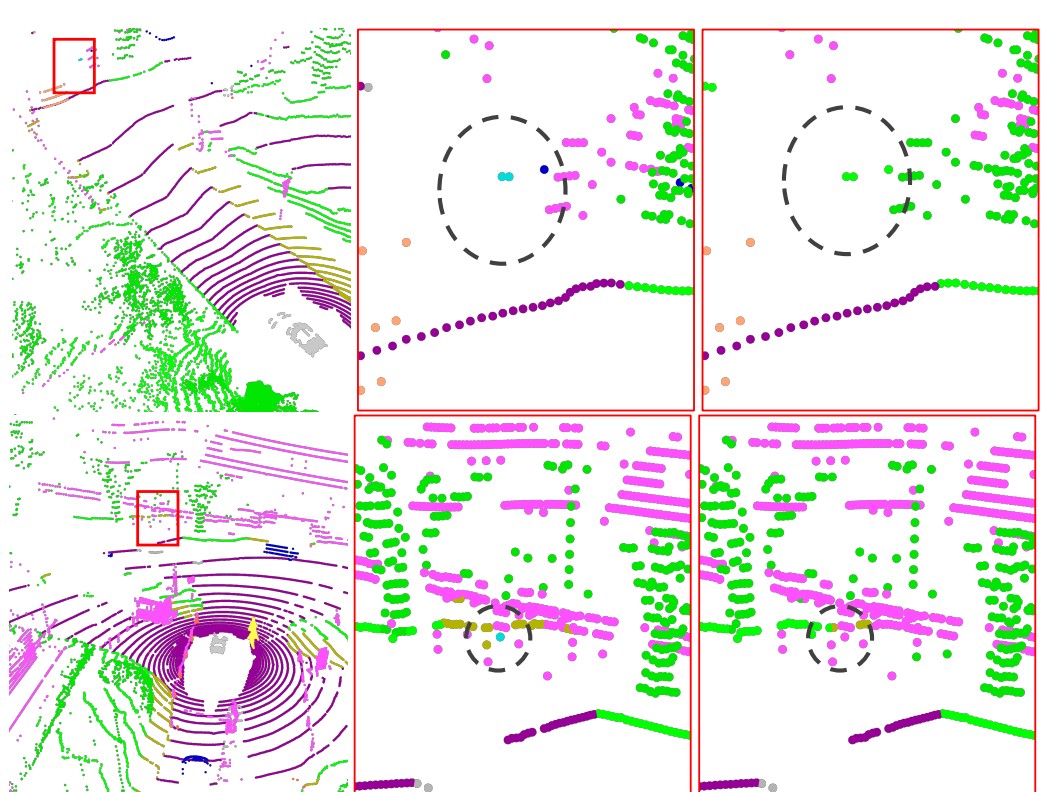

Figure 12: Visualization of Failure Cases . Models are trained on nuScenes with 1% labels. Columns from left to right show the scene overview, ground-truth annotations, and model predictions. Points colored in light blue indicate those labeled as *bicycle*.

Table 20: Comparison of ensemble performance across different model-training configurations on nuScenes and SemanticKITTI at varying label ratios.

| Ensemble Method | nuScenes | | | SemanticKITTI | | |
|---|---|---|---|---|---|---|
| | 1% | 20% | 50% | 1% | 20% | 50% |
| Direct Ensemble | 52.3 | 67.8 | 71.5 | 40.7 | 56.8 | 60.5 |
| DML Zhang et al. (2018) | 40.6 | 50.8 | 66.6 | 42.2 | 49.1 | 58.9 |
| *CoLLiS* (w/o Ensemble) | 63.2 | 74.2 | 75.8 | 56.0 | 64.9 | 66.2 |
| *CoLLiS* | **64.5** | **75.5** | **76.2** | **57.6** | **66.4** | **66.8** |

