# OpenReview forum: "Beyond Dual Representations: Collaborative Learning for Semi-Supervised LiDAR Semantic Segmentation"
_ICLR.cc/2026/Conference — Submitted to ICLR 2026_

### Official Review · Reviewer_qjPV · 2025-10-25

**Soundness:** 3
**Presentation:** 3
**Contribution:** 3
**Rating:** 8
**Confidence:** 4

**Summary:**

The manuscript introduces CoLLiS, a collaborative learning framework for semi-supervised LiDAR semantic segmentation. Unlike traditional two-stage SemiSL methods that rely on a single representation and pseudo-label distillation, or recent dual-representation approaches with cross-view training, CoLLiS treats multiple LiDAR representations (range-view, voxel, polar, etc.) as coequal students trained in a single stage.

The key ideas of this work are:
1. consensus-driven augmentation that adjusts perturbation strength based on inter-student agreement.
2. adaptive pseudo-labeling and distillation balancing absolute and relative reliability among models.
3. post-training ensemble to further consolidate multi-representation knowledge.

Experiments across nuScenes, SemanticKITTI, and ScribbleKITTI show consistent improvements over state-of-the-art SemiSL baselines, especially under scarce-label settings. Ablations further highlight the contributions of adaptive reliability measures, dynamic augmentation, and multi-representation collaboration

**Strengths:**

(+) The manuscript addresses the inherent confirmation bias in SemiSL for LiDAR segmentation and frames it in the context of representation diversity, which is timely and practically important for autonomous driving. Moving beyond dual representation methods (e.g., IT2), CoLLiS generalizes the paradigm to multiple representations in a unified single-stage pipeline, which is conceptually neat and reduces inefficiencies of prior two-stage designs.

(+) The introduction of consensus-driven augmentation and adaptive reliability-based distillation are methodologically sound extensions that directly target SemiSL challenges. The use of both absolute and relative reliability to weigh pseudo-labels is particularly well-justified.

(+) Results across three benchmarks and multiple label ratios demonstrate consistent gains, with notable improvements under 1% and 10% labels. Ablations are comprehensive, showing the additive benefits of each design choice, and efficiency analysis further supports the single-stage design.

**Weaknesses:**

(-) Limited exploration of representation scalability. Although the framework is presented as scalable to multiple representations, most experiments rely on three specific ones (frustum-range, polar, voxel). The performance impact of adding/removing representations is analyzed, but the exploration of “how far” this scalability goes (e.g., with more than three representations, or novel hybrid encoders) is limited.

(-) While the manuscript introduces several well-motivated components, each (consensus-driven augmentation, online collaborative distillation, post-training ensemble) is individually incremental relative to existing SemiSL and CoL frameworks. The novelty mainly lies in their combination for LiDAR segmentation rather than in fundamentally new algorithmic ideas.

(-) Several components (pseudo-labeling, distillation, ensemble) hinge on confidence measures, which are known to be brittle under distribution shift. While effective in benchmarks, the generalizability to real-world OOD or sensor-noise scenarios is not fully demonstrated, for example, Robo3D.

(-) aluation comprehensiveness: Please consider comparing FRNet [Xu, et al.] and LarseMix++ [Kong, et al.] for a more comprehensive evaluation on the semi-supervised LiDAR segmentation benchmark. Additionally, there are several self-supervised LiDAR segmentation methods that are evaluated on the same benchmark; please consider including those as well, such as SLidR [Sautier, et al.], Seal [Liu, et al.], and SuperFlow [Xu, et al.].

References:

- FRNet [Xu, et al.] FRNet: Frustum-Range Networks for Scalable LiDAR Segmentation. TIP, 2025.
- LarseMix++ [Kong, et al.] Multi-Modal Data-Efficient 3D Scene Understanding for Autonomous Driving. TPAMI, 2025.
- SLidR [Sautier, et al.] Image-to-Lidar Self-Supervised Distillation for Autonomous Driving Data. CVPR, 2022.
- Seal [Liu, et al.] Segment Any Point Cloud Sequences by Distilling Vision Foundation Models. NeurIPS, 2023.
- SuperFlow [Xu, et al.] 4D Contrastive Superflows are Dense 3D Representation Learners. ECCV, 2024.

---

**Minor Suggestions**

(-) Please expand discussion of potential weaknesses in confirmation bias mitigation: for example, do disagreements among weak students introduce noise despite adaptive weighting?

(-) Include visualizations or failure cases for extremely rare long-tail classes (beyond bicycles), and further discuss why collaboration does not fully overcome extreme imbalance.

(-) Clarify the computational trade-offs more directly: while training efficiency is reported, inference cost under ensemble use could be highlighted.

(-) Improve figure readability (e.g., small font sizes in Fig. 3 and Fig. 4) for clarity.

**Questions:**

The paper is well-motivated, tackling the important issue of confirmation bias in SemiSL for LiDAR segmentation, and presents a clean extension from dual to multi-representation collaborative learning. The empirical results are strong and broadly convincing across multiple benchmarks, particularly in scarce-label settings. However, the technical novelty is a little bit moderate, as the main contributions combine existing ideas (collaborative learning, confidence-based pseudo-labeling, augmentation adaptation) rather than introducing a fundamentally new principle. The improvements at higher label ratios are incremental, which tempers the overall impact.

That said, the manuscript is solidly executed, well-written, and relevant to the ICLR community, and it could be of interest given the growing focus on SemiSL and multi-representation learning for 3D perception. With minor refinements and broader baseline coverage, this work has potential for acceptance. With additional efforts in addressing the main weaknesses and minor suggestions (as detailed above), the manuscript could have a strong impact.

---

> ### Author Response · Authors · 2025-11-21
> **Response to Reviewer qjPV (Part 1)**
>
> We thank the reviewer for the positive feedback and thoughtful suggestions. We appreciate your recognition of the contribution of this work to the community. Below we provide our responses to the raised points, and we hope they adequately address your concerns. We will also revise the paper to incorporate the discussed points, and we welcome any further questions.
>
> >W1: Framework scalability with more than three representation.
>
> To verify the scalability of our framework, we include an additional representation (raw points) and adopt Point Transformer v3 (PTv3; Wu et al., 2024) as its backbone. For a clear comparison, we report the performance trained with different number of representation. All participants are evaluated standalone on validation set. It clearly show that the performance of stronger representations (F-Range and Voxel) saturates while the underperforming representation (Polar) benefits substantially from adding one more representation for collaboration.
>
> **nuScenes**
> | mIoU (1%) | FRNet (F-Range)| PolarNet (Polar)|Cylinder3D (Voxel)| PTv3 (Raw points)|
> | -------- | ------|-------|-------|-------|
> |#Models = 2 |  62.5 | -  |59.2  | -  |
> |#Models = 2 |  60.8  | 55.2  |-  | - |
> |#Models = 3 |  63.2  | 57.9  | 61.1  |-  |
> |#Models = 3 |  63.5  | -  |61.3  | 60.7  |
> |#Models = 4 |  63.4  | 58.6  |61.6  |61.2  |
>
> **SemanticKITTI**
> | mIoU (1%) | FRNet (F-Range)| PolarNet (Polar)|Cylinder3D (Voxel)| PTv3 (Raw points)|
> | -------- | ------|-------|-------|-------|
> |#Models = 2 |  55.1 | -  | 51.8 | -  |
> |#Models = 2 |   53.9 | 45.0  |-  |-  |
> |#Models = 3 |  56.0  | 46.8  | 53.2  |-  |
> |#Models = 3 |  56.8  | -  |  54.3   | 54.8  |
> |#Models = 4 |  56.8  | 48.9  | 54.7  | 55.2  |
>
> An additional note:  With only 1% labels, PTv3 performs poorly compared to other representation (See the table below).  Yet, when integrated into our collaborative framework, its performance improves substantially. This shows that the gain stems from collaboration instead of backbone strength.
>
> **nuScenes (supervised)**
>
> | mIoU  | FRNet (F-Range)| PolarNet (Polar)|Cylinder3D (Voxel)| PTv3 (Raw points)|
> | -------- | ------|-------|-------|-------|
> |1% |  51.9 | 46.5  |50.9  | **48.0**  |
> |10% |  68.1  | 58.5  |65.9  |**67.8**  |
> |20% |  70.9  |  63.9 |66.6  |**71.5**  |
> |50% |  74.6  | 68.4  |71.2  |**75.9**  |
>
>
>
> >W2: Incremental Novelty
>
> We would like to clarify that our contribution extends beyond a simple combination of existing idea:
> 1. No prior SemiSL methods scale beyond two representations. This is a breakpoint as conventional frameworks are limited by its two-stage design and fixed distillation paradigm (weak->strong augmentation).
> 2. Although Collaborative Learning (CoL) can effectively address this scalability bottleneck, existing CoL methods operate in supervised image-domain settings and does not work under label-scarce LiDAR settings. To overcome this limitation, CoLLiS introduces carefully designed dynamic augmentation mechanism, reliability modeling and pseudo-labeling strategy, to enable CoL's capability in LiDAR semi-supervised learning.
>
> Thus, our contribution is not a direct transfer or combination of existing concepts, but a problem-driven (scalability) extension that enables collaborative learning to operate in a new, significantly more challenging scenario.
>
>
> >W3: OOD Generalizability
>
> We provide the results on Robo3D (nuScenes-C) below. We follow the same protocol and report mean corruption error (mCE) and mean resilience rate (mRR). We will include the full evaluation table covering all corruption types in the revised version of the paper.
>
> **nuScenes-C**  Semantic Segmentation
>
> | Method  | mCE &#8595; | mRR &#8593;| mIoU &#8593;|
> |-------|-------|-------|-------|
> PolarNet| 115.1| 76.3 | 71.4|
> PolarNet*| 111.5| 77.9 | 74.3|
> |||||
> Cylinder3D| 111.8| 72.9 | 76.2|
> Cylinder3D*| 110.1| 74.1 | 76.8|
> |||||
> FRNet| 98.6| 77.5 | 77.7|
> FRNet*| 99.8| 77.0 | 77.1|
>
> *: Trained with CoLLiS

---

> ### Author Response · Authors · 2025-11-21
> **Response to Reviewer qjPV (Part 2)**
>
> >W4: Missing several benchmarks for comparison
>
> We appreciate the reviewer’s suggestions. However, the proposed methods operate under settings that are not directly comparable to our problem formulation.
>
> SLidR exploits image–LiDAR correspondences, LaserMix++ introduces both image and language guidance, and SuperFlow and Seal further leverage temporal cues from LiDAR sequences. While these methods are effective, they operate under different supervision settings by incorporating additional modalities or temporal signals beyond the single-frame LiDAR inputs used in our experiments. Therefore, direct comparison would not be fair. Our work focuses on the single-modality, single-frame LiDAR SemiSL setting, which is consistent with other prior works such as IT2 and LaserMix.
>
> However, we acknowledge that these directions are orthogonal to our method and represent promising avenues for future extension. To improve clarity, we will add a section discussing limitations and future works, and include the following text:
>
> >>*While effective, our framework does not include multi-modality or temporal cues for training. Integrating collaborative learning with richer supervision sources, e.g. image-LiDAR alignment, language guidance and temporal consistency [**References**] may further enhance label-efficient training in real-world autonomous driving applications. We leave the integration of these complementary signals into our framework as valuable directions for future exploration.*
>
>
>
> >Minor W1: Please expand discussion of potential weaknesses in confirmation bias mitigation: for example, do disagreements among weak students introduce noise despite adaptive weighting?
>
> We agree that disagreements among weaker students may introduce noise. Such conflicts naturally occur because we intentionally avoid enforcing hard consensus in order to reduce confirmation bias. Our adaptive reliability weights softly resolve these conflicts rather than eliminating them entirely. The effectiveness of this weighting mechanism is validated in Table 12. We will expand this discussion in the revised manuscript to improve clarity.
>
>
> > Minor W2: Include visualizations or failure cases for extremely rare long-tail classes (beyond bicycles), and further discuss why collaboration does not fully overcome extreme imbalance.
>
> We thank the reviewer for the suggestion. We will include such visualization in the revised manuscript.
>
> > Minor W3: Clarify the computational trade-offs more directly: while training efficiency is reported, inference cost under ensemble use could be highlighted.
>
> That is a very good suggestion. We will report the inference cost for both ensemble and single-model use to provide a clearer comparison.
>
>
> > Minor W4: Improve figure readability (e.g., small font sizes in Fig. 3 and Fig. 4) for clarity.
>
>
> We have enlarged the font sizes and improved the layout in Fig. 3 and Fig. 4 to enhance readability. These changes are already reflected in the revision.

---

> > ### Comment · Reviewer_qjPV · 2025-11-27
> >
> > Thanks to the authors for the responses. I have read the responses and other reviewers' comments. Most concerns raised from my side have been well addressed. Therefore, I am leaning to maintain the postive rating and recommend towards acceptance.

---

> > > ### Author Response · Authors · 2025-11-27
> > >
> > > We sincerely appreciate the reviewer’s decision to maintain a positive rating and the recommendation toward acceptance. We truly thank the reviewer for the valuable feedback, which has contributed meaningfully to improving the quality of our work.

---

### Official Review · Reviewer_5ZQn · 2025-10-25

**Soundness:** 3
**Presentation:** 3
**Contribution:** 2
**Rating:** 4
**Confidence:** 3

**Summary:**

This paper tackles the challenge of semi-supervised semantic segmentation for LiDAR point cloud data. The authors highlight key limitations in existing two-stage student–teacher frameworks, particularly their reliance on unidirectional knowledge distillation from a single teacher model. This setup can lead to confirmation bias, where erroneous pseudo-labels are repeatedly reinforced during training. Furthermore, extending such two-stage methods to multi-view LiDAR representations introduces significant complexity and computational overhead. To address these issues, the paper introduces a teacher-free collaborative learning framework that distills knowledge directly from student networks using a carefully crafted pseudo-labelling strategy. The approach is further strengthened by a novel data augmentation technique and a post-training ensemble mechanism, both designed to boost model performance. Extensive experiments on nuScenes, SemanticKITTI, and ScribbleKITTI datasets demonstrate that the proposed method consistently outperforms baseline models.

**Strengths:**

- The paper is generally well-organized and easy to follow.
- The idea of adopting a single-stage framework, along with the motivation grounded in curriculum learning, is interesting and well-motivated.
- The proposed method show better training efficiency compared with the baseline methods.
- The paper have provied extensive experiments to demonstrate the effectiveness of the proposed method.

**Weaknesses:**

- The authors may consider clarifying what $M_1$, $M_2$, and $M_3$ represent in Figure~1. It is unclear whether these refer to specific LiDAR representations, model architectures, or transformation outputs. A brief explanation in the figure caption or main text would improve clarity.
- Is it possible that the predictions are biased toward a single modality, potentially limiting the benefits of cross-representation collaboration?
-Eq.7 relies on filtering pseudo-labels based on the confidence score $c(P_{s1})$. Could the authors provide an analysis of how the coverage of confident predictions changes from early to late training stages? Specifically, how much of the unlabeled data is retained or discarded over time due to the dynamic threshold?
- In Table. 9, the performance of CoLLiS-Hete appears to be lower than baseline methods such as IT2 and LaserMix under certain settings. I would appreciate it if the authors could elaborate on the underlying reasons for this discrepancy. Specifically, it would be helpful to understand whether architectural incompatibility, representation redundancy, or training instability might have contributed to the observed gap.

**Questions:**

See Weaknesses section.

---

> ### Author Response · Authors · 2025-11-21
> **Response to Reviewer 5ZQn**
>
> We are glad that the reviewer found our idea is interesting and confirm the overall presentation quality of our paper, and also appreciate reviewer's valuable feedbacks. We address the raised concerns as follows:
>
> > W1: Clarifying what M1, M2 and M3 represent in Fig.1.
>
> We have added explicit definitions of M1, M2, and M3 in the caption of Fig. 1 (highlighted in orange).
>
> > W2: Potential Modality Bias in Confidence-Based Pseudo-Labeling
>
> We thank the reviewer for raising this point. We have added a figure in the revised version (See Figure 10(a) in Appendix). The plot shows that each representation retains a certain portion of pseudo-labels after thresholding, and the overall rate increases progressively as training proceeds. This supports our hypothesis that pseudo-label reliability improves over time. While stronger representations tend to retain a higher proportion, weaker ones are not completely discarded. Our thresholding process is done independently for each student and this allows even weaker models to retain valid pseudo-labels in regions where they are confident. This ensures that collaboration remains active across views rather than collapsing to a single dominant modality.
>
> Moreover, from consistent improvements of all representations shown in Table 1, it also confirms that the system does not rely solely on the strongest architecture, and all models collaborate actively during training. If they bias toward on knowledge from the strongest architecture,  collaborative learning would not be effective, as the strongest model would stop improving due to limited feedback from other weaker peers.
>
>
> > W3: Performance gap of CoLLiS-Hete to IT2/LaserMix under certain settings.
>
> We thank the reviewer for this insightful question. First, we do not observe distinct instability during training (loss converges normally). The lower performance of CoLLiS-Hete in Table 9 primarily stems from limitations of the range-view representation. As discussed in Section 4.4 (*Impact of representation choice*), range-view is derived via spherical projection from 3D space, which inherently introduces irreversible information loss when multiple points collapse onto the same grid cell. This loss cannot be fully recovered through collaboration unless other views provide compensatory geometric detail. This explanation is further supported by the fact that IT2-Hete also experiences a notable decline relative to its dual-representation baseline.
>
> A secondary factor is the comparatively weaker performance of transformer-based architectures (e.g., RangeViT) under extreme label scarcity, such as low annotation ratios or sparse annotations like ScribbleKITTI. With insufficient labeled samples, the model fails to learn strong geometric inductive biases (Ando et al. (2023),  Dosovitskiy et al. (2021)). Such weakness can propagate pseudo-label noise and restrict the effectiveness of collaboration.

---

### Official Review · Reviewer_A5B5 · 2025-11-02

**Soundness:** 3
**Presentation:** 3
**Contribution:** 1
**Rating:** 4
**Confidence:** 4

**Summary:**

This paper proposes CoLLiS, a semi-supervised LiDAR semantic segmentation framework that integrates multi-representation collaborative learning, consensus-driven augmentation, and adaptive pseudo-label distillation. The method aims to alleviate confirmation bias and improve model generalization. Experiments on three benchmarks — nuScenes, SemanticKITTI, and ScribbleKITTI — demonstrate that CoLLiS outperforms existing semi-supervised approaches under low-annotation settings.

**Strengths:**

1. Proposes a novel single-stage multi-representation collaborative framework that attempts to mitigate confirmation bias in semi-supervised LiDAR segmentation.
2. Conducts extensive experiments across multiple datasets and label ratios, showing consistent performance improvements.
3. Introduces a consensus-driven augmentation (CDA) and adaptive pseudo-labeling mechanism that demonstrate a degree of methodological innovation.

**Weaknesses:**

1. Unsubstantiated claim of “confirmation bias mitigation.” The paper repeatedly claims that CoLLiS alleviates confirmation bias（line 25）, yet no quantitative definition or measurable evidence is provided. There is no analysis of pseudo-label precision/recall, confidence distribution, or inter-model consistency to verify that bias indeed decreases. Merely reporting an mIoU improvement does not prove bias reduction. From a learning-theory perspective, confirmation bias refers to self-reinforcing model errors under pseudo-labeling. Without metrics or curves showing how pseudo-label quality evolves, the claim remains unsupported and unverifiable.
2. Lack of empirical proof of “confirmation bias can be amplified when multiple representations produce inconsistent pseudo-labels.”(line 231) While the method assumes that multi-representation learning provides complementary information, the paper does not empirically verify such synergy. Specifically, it omits: 1）Pairwise pseudo-label agreement or error-overlap matrices between representations; 2）Comparison between “independent training + ensemble” and “online collaborative training”; 3）Analysis of whether one model can compensate when another degrades (i.e., recovery behavior). Therefore, the supposed collaboration effect remains anecdotal rather than demonstrated.
3. Weak theoretical support and no convergence or bias-correction analysis. The paper provides no theoretical explanation of the optimization objective, no analysis of pseudo-label distribution convergence, and no discussion of the method’s upper bound or failure cases. Without such analysis, the work stays at the empirical stacking level and lacks a clear learning principle.
4. Limited improvement on long-tailed classes. Figure 6 shows that under the 1% labeled regime, the bicycle class IoU remains extremely low (≈3%), indicating that the approach still fails to handle severely imbalanced or rare classes effectively, despite overall mIoU gains.
5. Pseudo-label reliability is not quantified. The adaptive pseudo-label weighting depends on model confidence, but the paper does not provide curves of pseudo-label accuracy during training or any calibration analysis. It is unclear whether pseudo-labels actually become more accurate as training progresses.
6. Reproducibility concerns.The code is not released, and critical hyperparameters (e.g., \delta_0 and \lambda_0) lack justification or tuning methodology. Without implementation details or theoretical grounding, it is difficult for readers to reproduce or validate the results.

**Questions:**

Please refer to the weaknesses for my main concerns.
In addition, I would like the authors to clarify the following:
1. Definition and measurement: How do the authors define confirmation bias quantitatively in this context, and what concrete evidence (e.g., pseudo-label precision/recall curves, confidence calibration, or inter-model disagreement) supports that CoLLiS actually mitigates it?
2. Collaborative effect validation: Can the authors provide ablation studies comparing (a) independently trained models with ensemble inference and (b) online collaborative training, to demonstrate that real cross-model synergy exists?
3. Reliability of adaptive pseudo-labeling: How does pseudo-label accuracy evolve during training? Have the authors evaluated the correlation between confidence weighting and true pseudo-label correctness?
4. Long-tail robustness: What mechanisms in CoLLiS specifically target rare or long-tailed categories? Could class-balanced pseudo-label filtering or re-weighting further improve performance on rare classes like bicycle?
5. Theoretical grounding: Does the optimization objective of CoLLiS guarantee convergence or stability? Can the authors provide at least empirical loss-curve evidence that collaborative training remains stable?
6. Hyperparameter rationale: Will the code be released? On what theoretical or empirical basis were \delta_0 and \lambda_0 chosen, and how sensitive is the performance to these values?

---

> ### Author Response · Authors · 2025-11-21
> **Response to Reviewer A5B5 (Part 1)**
>
> We appreciate the reviewer’s constructive suggestions, which will greatly enhance the clarity of our work. We are also pleased that the reviewer acknowledges the novelty of our work. We address the concerns below.
>
> >W1: Lack of measurable evidence on “confirmation bias mitigation.”
>
> We thank the reviewer for the constructive suggestion and agree that evidence beyond final performance is needed to support our claim on confirmation bias mitigation. To address this, we add quantitative analyses tracking pseudo-label behavior during training.
>
> (1) Confirmation Bias Metric:
>
> Following prior works in studying pseudo-labels in image domain (Arazo et al. (2020)), we measure the certainty of incorrect predictions, which is adapted to LiDAR segmentation:
>
> $$
> r = -\frac{1}{N^{\star}} \sum_{n=1}^{N^{\star}} \mathcal{U}^T \log\big(P^{\star}),
> $$
> where $N^{\star}$ is the number of incorrect pseudo-labels, $P^{\star}$ is their predicted probability distribution and $\mathcal{U}$ is a uniform prior. This metric evaluates how confident the model is when it is wrong. Higher confidence on incorrect predictions indicates stronger confirmation bias, as errors are reinforced during self-training. Lower values indicate reduced bias.
>
> We report results in Figure 11(b). For comparison, we include standard collaborative training (DML, Zhang et al., 2018) under identical settings. Results show that:
>
> - *DML progressively becomes overconfident on incorrect samples.*
> - *CoLLiS consistently suppresses confidence in incorrect predictions throughout training.*
>
> (2) Pseudo-label Quality
>
> We additionally report pseudo-label accuracy over training (Fig. 10(b)), again compared with DML. Our method produces higher-quality pseudo-labels throughout training.
>
> **To summarize, these analyses provide quantitative evidence that CoLLiS mitigates confirmation bias by reducing overconfidence on incorrect predictions and improving pseudo-label quality.**
>
> >W2:  Lack of empirical proof on synergy of collaboration
>
> We thank the reviewer for the thoughtful comments. We address the three requested aspects below.
>
> (1) Pairwise pseudo-label agreement across representations:
>
> As shown in Fig. 10(a), pseudo-labels are maintained independently for each representation at every training step, which makes direct measurement of pairwise agreement non-trivial. Instead, we report pseudo-label accuracy and retention rate per representation in Fig. 10(b) and (a). All representations quickly achieve high accuracy in their pseudo-labels (>80% accuracy after ~2% of training, while generating increasing amount of valid pseudo-labels. This indicates that collaborative learning effectively constrains error propagation in pseudo-labels.
>
> (2) Comparison to independent training + ensemble:
>
> Please refer to our response to W1 of Reviewer J17C.
>
> (3) Recovery ability when one model degrades
>
> Fig. 11(a) tracks performance (mIoU on validation set) throughout training. Under standard collaborative learning (DML), performance collapses mid-training (~40%) and continues to decline for all models, indicating no recovery from degradation. In contrast, CoLLiS maintains stable improvements across all representations throughout training.
>
>
> Additional evidence of synergy:
>
> We highlight that all models improve simultaneously in Table 1. If collaboration relied solely on knowledge from the strongest architecture, weaker models would benefit while the strongest would plateau. Yet, we observe improvements across all representations. This behavior validates that all models collaborate actively during training rather than rely solely on the strongest one.
>
>
> >W3: The paper provides no theoretical explanation of the optimization objective, no analysis of pseudo-label distribution convergence, and no discussion of the method’s upper bound or failure cases.
>
> We appreciate the reviewer’s feedback. Regarding pseudo-label distribution, please refer to our response to Q2, where we provide quantitative results of pseudo-label retention and accuracy over training.
>
> For upper-bound performance, please refer to our response to W1 of Reviewer qjPV, where we further explore scalability by extending the framework to four representations. While it shows slight improvement, the performance of each representation tends to saturate.
>
> Regarding failure case, we have discussed the limitation in Section 4.4 that our model struggles with extremely rare long-tail classes. We also clarify this point in our response to W4 and Q4.

---

> ### Author Response · Authors · 2025-11-21
> **Response to Reviewer A5B5 (Part 2)**
>
> >W4: Limited improvement on extremely rare classes.
>
> We acknowledge this limitation and have discussed it in Section 4.4. To elaborate the issue more, we provide the supervised performance on the bicycle class below. All participating representations suffer from extreme class imbalance with 1% labels, resulting in very low IoU scores. This suggests that the bottleneck arises from intrinsic data scarcity rather than the collaborative paradigm. Therefore, it is difficult for CoLLiS to bridge the gap without additional class-rebalancing strategies. Nevertheless, CoLLiS still yields consistent improvements over individual representations and other SemiSL baselines on top-rare classes, as shown in Figure 6(a\).
>
> **nuScenes 1% labels**
> | IoU bicycle | FRNet (F-Range)| PolarNet (Polar)| Cylinder3D (Voxel)|
> | -------- | ------|-------|-------|
> | sup.only | 1.0  | 0.9  | 1.9  |
> | Ours |  3.1 | 2.1  | 3.9|
>
>
>
> >W5: Pseudo-label reliability is not quantified.
>
> Please refer to our response to Q2
>
> >W6: Reproducibility concerns
>
> Please refer to our response to Q6.
>
>
>
> >Q1: Definition and measurement: How do the authors define confirmation bias quantitatively in this context, and what concrete evidence (e.g., pseudo-label precision/recall curves, confidence calibration, or inter-model disagreement) supports that CoLLiS actually mitigates it?
>
> In our paper, we define confirmation bias as that models can reinforce to incorrect pseudo-labels during training under semi-supervised setting (as illustrated in line 074 in paper). We reflect the effectiveness of CoLLiS for mitigating confirmation bias by observing the performance on tail classes, as they are significantly vulnerable to overconfidence toward non-long-tail classes due to substantially fewer annotations. Across both rare and extremely rare categories, our method show substantial improvements over state-of-the-art methods.
>
> However, we understand and acknowledge the reviewer's concern that more concrete evidence and diverse aspects are necessary for rigor explanation. Therefore, we take the reviewer's suggestion and additionally report following results to support our claim in the paper:
> 1. Certainty of incorrect pseudo labels, which quantitatively reflects the confirmation bias during training (details given in our response to W1)
> 2. Pseudo-label accuracy and retention rate after thresholding and filtering out invalid predictions.  (details given in our response to W2
>
> >Q2: Collaborative effect validation
>
> Please refer to our response to W1 of Reviewer J17C
>
> >Q3: Reliability of adaptive pseudo-labeling: How does pseudo-label accuracy evolve during training? Have the authors evaluated the correlation between confidence weighting and true pseudo-label correctness?
>
> Yes. As discussed in W2, we analyze how pseudo-label reliability evolves during training. Figure 10(a) reports the retention rate of pseudo-labels after confidence thresholding, while Figure 10(b) shows the accuracy of the retained pseudo-labels over time. We observe that pseudo-label accuracy remains consistently high while the retention rate increases as training progresses This indicates that the model preserves more reliable supervision signals over time. In the mean time, mIoU on validation set improves substantially (see Figure 11(a)). Overall, it supports that our mechanism based on confidence weighting is an effective proxy for generating high-quality pseudo-labels under the collaborative setup
>
>
> >Q4: Long-tail robustness: What mechanisms in CoLLiS specifically target rare or long-tailed categories? Could class-balanced pseudo-label filtering or re-weighting further improve performance on rare classes like bicycle?
>
> In Figure 6(a), we have integrated a long-tail class–pasting strategy to rebalance rare categories, however, the improvement remains minor under the 1% label regime because all representations receive extremely sparse supervision for these categories (as discussed in W4).
>
> We therefore expect dataset-internal rebalancing strategies (e.g., re-weighting or class-balanced filtering) to provide only marginal improvements in this extreme setting. More substantial gains may require enriching rare-class samples through synthetic data, e.g., pseudo instance generation [a] or simulator-generated scenes [b], which we view as promising future directions.
>
> *Reference:*
>
> *a: Chang, Mincheol et al. “Just Add $100 More: Augmenting NeRF-based Pseudo-LiDAR Point Cloud for Resolving Class-imbalance Problem.” NIPS '24: Proceedings of the 38th International Conference on Neural Information Processing Systems*
>
> *b: Dosovitskiy, Alexey, et al. "CARLA: An open urban driving simulator." Conference on robot learning. PMLR, 2017.*

---

> ### Author Response · Authors · 2025-11-21
> **Response to Reviewer A5B5 (Part 3)**
>
> >Q5: Theoretical grounding: Does the optimization objective of CoLLiS guarantee convergence or stability? Can the authors provide at least empirical loss-curve evidence that collaborative training remains stable?
>
> Yes, we visualize an example of training loss curve in Figure 9. The loss decreases steadily throughout training.
>
>
> >Q6: Will the code be released? On what theoretical or empirical basis were \delta_0 and \lambda_0 chosen, and how sensitive is the performance to these values?
>
> Yes, the code will be released upon acceptance (as stated in the abstract). To ensure reproducibility, we have additionally specified all implementation details in Section B in Appendix, including datasets, hyperparameter settings, hardware configurations, and data augmentation protocols.
>
> For $\lambda_0$, we have provided the experimental results in C.3 and Table 11 in Appendix. For $\delta_0$, we do not include an ablation as its role is to restrict the reliance on pseudo-labels in early training. This aligns with standard practice in prior pseudo-labeling works (e.g., Wang et al., 2022), and we find performance to be stable as long as $\delta_0$ is sufficiently high initially.

---

### Official Review · Reviewer_J17C · 2025-11-03

**Soundness:** 3
**Presentation:** 3
**Contribution:** 2
**Rating:** 4
**Confidence:** 4

**Summary:**

This paper introduces CoLLiS, a collaborative learning framework for semi-supervised LiDAR semantic segmentation. Unlike dual-representation methods such as It Takes Two (IT2), CoLLiS leverages three distinct LiDAR representations (Frustum-Range, Polar, and Voxel) in a joint, single-stage setup, where all models are coequal learners. The key contributions include the CDA strategy and two dynamic distillation weights: AR and RR. These mechanisms aim to allow collaborative learning across diverse representations while mitigating confirmation bias. CoLLiS achieves superior performance over prior methods across three datasets (nuScenes, SemanticKITTI, ScribbleKITTI), particularly in low-label regimes.

**Strengths:**

1. As shown in Table 1, CoLLiS consistently outperforms all SOTA methods—including IT2—across the nuScenes, SemanticKITTI, and ScribbleKITTI benchmarks and under all label ratios. Notably, the performance gain is most pronounced in extremely low-label scenarios.
2. Despite utilizing three models, CoLLiS achieves faster training speed and lower GPU memory usage than IT2 (which employs only two models), as shown in Table 5. This demonstrates that the single-stage collaborative design of CoLLiS is highly efficient.
3. The long-tail class analysis in Figure 6 shows that CoLLiS significantly improves the performance of rare classes, which are most vulnerable to confirmation bias. This confirms that CoLLiS effectively addresses the central issue it targets.
4. The paper validates its mechanisms with extensive experiments.
	- The main Ablation Study clearly demonstrates that each component of CoLLiS (AR, RR, CDA, and +Polar) contributes statistically significant improvements to the overall performance.
	- Appendix experiments (C.3, Table 12) prove that the adaptive AR/RR scheme is a core contribution, significantly outperforming a fixed-weight approach (by +2.1pp at 1% labels).
	- Appendix C.1 (Table 9) also highlights the framework's flexibility, showing performance gains even when collaborating different architectures (CNN+ViT) on a single representation.

**Weaknesses:**

1. The core idea, a multi-representation, multi-model semi-supervised framework, amounts to a form of learned ensemble training. The use of multiple representations and cross-model consistency is not conceptually new and resembles straightforward model ensembling with inductive bias diversity. The paper lacks a clear distinction between CoLLiS’s collaborative learning and post-hoc ensembling of independently trained models. Notably, the empirical analysis does not isolate whether CoLLiS's gains are intrinsic to joint training or could be achieved by independently training each model and ensembling their predictions.

2. While Table 5 reports that CoLLiS is more memory- and time-efficient during training compared to IT2, this comparison omits a critical consideration: inference-time cost. Since CoLLiS relies on three full LiDAR segmentation models at inference, the required compute and memory footprint is substantially higher than most single-model or dual-representation systems.
This raises concerns about practical deployability in autonomous driving, where low-latency and resource-constrained inference is a critical requirement. The paper does not address how CoLLiS could be compressed, distilled, or otherwise optimized for real-time use.

3. IT2 is described as a “two-stage” method, but this characterization appears inaccurate. Like CoLLiS, IT2 also performs end-to-end learning with dynamic pseudo-label generation. This misframing weakens the methodological comparison.

4. Conflated Contributions: The paper's core contribution (AR/RR collaboration) and IT2's core contribution (GMM contrastive learning) appear to be orthogonal. The paper does not evaluate CoLLiS with GMM, making it unclear whether the gains are from the CoLLiS framework, or simply from using more models (3 > 2).

5. Missing Recent Baselines: While the baselines are relatively recent, for an ICLR 2026 submission, the paper is missing comparisons to more recent SOTA methods from 2024 and 2025 (e.g., DDSemi (CVPR 2024), HiLoTs (CVPR 2025)). Including these would strengthen its SOTA claims.

6. Questionable Scalability: The paper justifies using three models as a “balanced trade-off” (Table 6) and does not explore N > 3. This implies that for N > 3, the training cost likely outweighs the performance gains, suggesting a practical scalability limit.

7. (Minor) On page 8, Table 4 the citation for IT2 is incorrectly listed as Kong et al. (2023b) (LaserMix). It should be Liu et al. (2025).

**Questions:**

1. The reproduced IT2 performance in Table 1 is significantly lower than the results reported in the original IT2 paper. Could the authors specify the exact experimental differences (e.g., backbone, resolution) that led to this discrepancy and justify the fairness of the SOTA comparison?
2. The performance and efficiency gains may stem from using the FRNet backbone rather than FIDNet. Could the authors provide a dual-representation ablation comparing CoLLiS (FRNet, Voxel) vs. CoLLiS (FIDNet, Voxel) to isolate the framework's true advantage? Furthermore, are the efficiency gains (Table 5) also a result of this backbone difference?
3. The core mechanisms of CoLLiS (AR/RR collaboration) and IT2 (GMM-based contrastive learning) do not appear to be mutually exclusive. Have the authors considered incorporating IT2’s GMM-based contrastive loss into the CoLLiS framework?

---

> ### Author Response · Authors · 2025-11-21
> **Response to Reviewer J17C (Part 1)**
>
> We thank the reviewer for constructive feedbacks. Our responses to the raised concerns are provided below:
> >W1: Are CoLLiS gains due to collaboration or ensembling?
>
> While CoLLiS benefits from multiple representations, its core contribution lies in joint collaborative training rather than post-hoc ensembling. We note that all reported results (except Table 3) are obtained with single-model inference. The performance gains are not due to ensemble aggregation.
>
> To isolate the effect of collaboration, we additionally (i) independently train each model and ensemble their predictions, and (ii) apply standard collaborative learning (DML, Zhang et al. (2018) ) without our proposed components, and then ensemble predictions accordingly. As shown in Table below, CoLLiS consistently outperforms both alternatives, and the improvements remain substantial even without ensemble use. This indicates that gains stem from our collaborative setup with reliability modeling and mutual distillation.
>
> We will include these results and add the corresponding discussion in the revised version of the paper.
>
> **nuScenes**
> | Ensemble Method | 1%| 20%|50%|
> | -------- | ------|-------|-------|
> | Direct Ensemble|  52.3 | 67.8 | 71.5|
> | DML, Zhang et al. (2018)|  40.6 | 50.8| 66.6 |
> | Ours |  64.5  | 75.5  | 76.2  |
> | Ours (w/o Ensemble) |  63.2 | 74.2 | 75.8 |
>
> **SemanticKITTI**
> | Ensemble Method | 1%| 20%|50%|
> | -------- |------|-------|-------|
> | Direct Ensemble |  40.7 | 56.8  | 60.5  |
> | DML, Zhang et al. (2018)|  42.2  | 49.1  | 58.9 |
> | Ours |  57.6  | 66.4  | 66.8 |
> | Ours (w/o Ensemble) |  56.0  | 64.9 | 66.2 |
>
> >W2: Inference Efficiency and Real-Time Deployability
>
> We clarify that CoLLiS does not require multi-model ensemble inference. All main results in Table 1 are obtained using single-model inference, so the reported performance gains do not incur additional inference cost. **We treat the ensemble as an optional deployment strategy rather than a requirement.** For clarity, we will make these points explicit in the revised manuscript and include inference-time cost for both single-model and ensemble configurations.
>
> Regarding deployability in real-time application, this capability is already demonstrated in our main paper (Table 4). Specifically, we treat the ensemble predictions from three collaboratively trained models as high-quality pseudo-labels and distill them into a lightweight model. This distilled model achieves performance close to the ensemble while being used standalone at inference. This significantly reduces computational overhead and enables real-time use.
>
> >W3: Mischaracterization of IT2’s Training Procedure
>
> We thank the reviewer for pointing this out. We acknowledge that IT2 is trained end-to-end with dynamically updated pseudo-labels. Our use of the term “two-stage” was not meant to suggest separate optimization phases, but rather to describe its **two-step forward pipeline** within each iteration: IT2 first performs a forward pass on weakly augmented inputs to generate pseudo-labels, and then a second forward pass on strongly augmented inputs for training. In contrast, CoLLiS uses a single forward pass per model and performs pseudo-labeling and distillation on-the-fly from the same predictions, which contributes to the efficiency gains reported in Table 5. The difference is visually illustrated in Figure 1(b\) and 1(c\) .
>
> To avoid confusion, we will revise the wording in the paper, replacing “two-stage” with a more precise description such as “two-stage forward pass with weak and strong augmentation” for IT2.
>
> >W4: Integration of IT2’s GMM Contrastive Learning
>
> As discussed in W3, the training pipeline of CoLLiS differs from IT2. Since IT2’s GMM modeling relies on embeddings extracted from weakly augmented inputs, directly integrating the GMM module into our framework is non-trivial.
>
> That said, we agree that such comparisons are valuable for a more comprehensive evaluation. To enable alignment with IT2’s setting, we instead incorporate our pseudo-labeling mechanism into the IT2 framework and report the corresponding results. Further details and experimental results are provided in our response to Q3.
>
> >W5: Missing Recent Baselines (e.g., DDSemi (CVPR 2024), HiLoTs (CVPR 2025))
>
> DDSemi (Li & Dong, 2024) is already included in our comparisons in Table 7. We did not include DDSemi in Table 1 because it uses a different data split for training, and mixing results across different protocols would lead to an unfair comparison.
>
> HiLoTs assumes multi-frame LiDAR sequences as input and leverages temporal cues to enhance learning, whereas our work focuses on the standard single-modality, single-frame SemiSL setting. Therefore, a direct comparison would not support a fair evaluation under the same supervision conditions. We view incorporating temporal cues as a promising future direction, and will include HiLoTs in the discussion of future works. Further details are provided in our response to W4 of Reviewer qjPV.

---

> ### Author Response · Authors · 2025-11-21
> **Response to Reviewer J17C (Part 2)**
>
> >W6: Framework scalability with more than three representation.
>
> Please refer to our response to W1 of Reviewer qjPV.
>
> >(Minor) W7: Incorrect citation in Table 4
>
> We appreciate the reviewer for identifying this mistake. The citation has been corrected in the revised manuscript.
>
> > Q1: Clarification on Reported IT2 Performance
>
> We thank the reviewer for raising this point. The performance of IT2 reported in Table 1 matches the results published in the original IT2 paper. We suspect the confusion may arise from interpreting the entries in the "Polar" block. The results labeled IT2-Range and IT2-Voxel do not correspond to the range-view or voxel representations themselves, rather, they refer to the polar representation trained using pseudo-labels together with range or voxel representations under IT2 settings, respectively. These numbers correspond to Table 5 in the IT2 paper.
>
> To ensure clarity, we will add a note in the revised manuscript to explicitly indicate that these rows refer to Polar trained via cross-view supervision, not range-view or voxel performance. We apologize for any confusion caused by the problem.
>
> If the reviewer was referring to a different discrepancy, we would be grateful if they could point us to the specific entries so we may address them fully.
>
>
> > Q2: Isolating framework gains from backbone differences
>
> While Table 6 presents results using Range + Voxel in a collaborative setting, we acknowledge that the comparison with IT2 is not explicitly highlighted in the current version.
>
> Motivated by the reviewer’s comment in W4, we additionally conduct an experiment that combines CoLLiS’s pseudo-labeling and distillation process with the IT2 setting. In IT2, pseudo-labels generated by one representation supervise the other in a unidirectional manner without any filtering process. In our variant, we instead use pseudo-labels from both representations for dual distillation, and apply our reliability modeling for pseudo-label filtering and adaptive weights to balance distillation.
>
> **nuScenes**
> | Range + Voxel | FIDNet 1%| FIDNet 10%| Cylinder3D 1%| Cylinder3D 10%|
> | -------- | ------|-------|-------|-------|
> | IT2|  56.5 | 71.2  | 57.5  | 72.1   |
> | CoLLiS-dual|  57.2  | 71.4  |  57.4 | 71.2 |
> | CoLLiS-dual + IT2 |  58.4  | 72.5  | 60.8  | 73.1 |
>
> Using the same representations as IT2, CoLLiS achieves comparable performance while requiring significantly less training time and memory. The gains here are smaller than those reported in Table 1 due to representation-specific limitations discussed in Section 4.4 (*Impact of Representation Choice*). Importantly, combining CoLLiS with IT2 yields performance that surpasses both individual methods, which indicates that two approaches are complementary.
>
> Regarding training efficiency, replacing FRNet with FIDNet results in 640 ms latency and 10.40 GB memory usage, which are even lower than the numbers reported in Table 5. This improvement stems from FIDNet’s use of a range-view representation and a lightweight backbone, making it more computationally efficient.
>
> We will incorporate this experiment and the corresponding discussion in the revised manuscript.
>
> >Q3: Have the authors considered incorporating IT2’s GMM-based contrastive loss into the CoLLiS framework?
>
> Yes, IT2’s GMM-based contrastive learning is complementary to our AR/RR collaborative framework and could potentially enhance cross-representation alignment. However, as discussed in W4, integrating GMM directly into CoLLiS is non-trivial due to difference in training procedure. In addition, maintaining prototypes (either via GMM or using memory banks (Li et al. (2023))) and computing contrastive losses for multiple representations substantially increases training time and memory usage. A key goal of this work is to design a scalable collaborative framework while keeping training cost manageable.
>
> We acknowledge that combining CoLLiS with contrastive learning is a promising direction, and we will include this discussion in the revised manuscript.

---

> > ### Comment · Reviewer_J17C · 2025-11-27
> > **post-rebuttal**
> >
> > The authors provided clear and detailed responses to my questions, and I appreciate the careful effort reflected in the rebuttal. Their additional analyses and experiments help strengthen several aspects of the submission, and I acknowledge the value of these clarifications.
> >
> > However, after considering the broader discussion across reviewers, some limitations remain. The conceptual novelty of the proposed approach appears limited, and the performance gains from increasing the number of representations are limited.
> >
> > In light of both the strengthened rebuttal and the remaining concerns, I will consider the overall contribution carefully before finalizing my score.

---

> > > ### Author Response · Authors · 2025-11-27
> > > **Follow-up clarifications to reviewer's response**
> > >
> > > We appreciate the reviewer’s follow-up and the acknowledgement of our strengthened rebuttal. Regarding the remaining concerns, we would like to clarify a few points:
> > >
> > > 1. *Contribution Scope and Significance:*
> > >
> > > The use of multiple LiDAR representations for semi-supervised LiDAR segmentation **remains underexplored**, primarily because existing SemiSL frameworks are constrained by their two-step design with fixed weak-to-strong distillation paradigm. These limitation have been noted in prior works but a practical solution has not been introduced.
> > >
> > > To bridge this gap, we introduce the new collaborative framework that **departs from prior two-step methodologies**. Our method enables scalable multi-representation collaboration and achieves state-of-the-art performance across various benchmarks and settings.
> > >
> > > Furthermore, beyond adopting the CoL concept, our method introduces **conceptually new components**, such as dynamic augmentation guided by peer agreement and adpative distillation weights with reliability modeling.
> > >
> > > We would also like to emphasize that our method (CoLLiS) is complementary to a variety of existing LiDAR SemiSL directions, including:
> > >  - Contrastive learning (as discussed in Q2)
> > >  - Long-tail class pasting augmentation (Figure 6(a))
> > >  - Enhanced frame downsampling strategies (Table 7)
> > >
> > > This compatibility suggests that CoLLiS can serve as a general, extensible framework rather than a single-method improvement, potentially broadening its utility and impact in the LiDAR community
> > >
> > >
> > > 2. *Significance of improvements*
> > >
> > > We argue that the improvements with increase of representations are **not limited**. First, scaling to additional representations yields consistent gains across all individual models, even under the most challenging evaluation setting we consider: **1% label ratio combined with a uniform data split**, which severely restricts scene diversity. Achieving consistent improvements under such conditions demonstrates the robustness of our collaborative mechanism.
> > >
> > > Second, we observe that gains appear on the weaker representations (e.g., Polar and Raw Points) are most significant. This pattern is particularly meaningful: **it shows that stronger representations are not dragged down but instead provide constructive guidance to weaker ones, demonstrating a key advantage of collaborative learning.** This supportive effect becomes more pronounced as the number of participating representations increases.

---

### Author Response · Authors · 2025-11-26
**Summary of Revisions in the Updated Manuscript**

Dear Reviewers,

We sincerely thank you for your efforts on reviewing our work and your constructive feedbacks. We have updated the manuscript accordingly. The major revisions are summarized below:

1. Additional evidence of collaboration
    - Added detailed discussion (Lines 396–406)
    - Included comparisons against independently trained ensembles and standard collaborative learning (Table 20)
    - Added analysis of recovery behavior (Figure 11a)
2. Additional evidence of confirmation bias mitigation
    - Introduced an explicit measurement for confirmation bias (Lines 424–431)
    - Added the corresponding plot (Figure 11b)
3. Convergence and distribution of pseudo-labels
    - Added discussion on pseudo-label dynamics (Lines 449–457)
    - Added pseudo-label retention curves (Figure 10a)
    - Added pseudo-label accuracy curves (Figure 10b)
4. Scalability analysis of the framework
    - Added discussion on scalability behavior (Lines 486–494)
    - Included results with 2, 3, and 4 collaborating models (Table 19)

5. Extended discussion of future directions (Lines 509–520)
6. Out-of-distribution robustness experiments (Appendix C.4)
7. Inference-time cost analysis (Appendix D.3)
8. Added plot of training loss curve (Figure 9)
9. Clarification of limitations on extremely long-tailed classes under low-label regimes
    - Added discussion (Appendix D.4)
    - Provided corresponding qualitative failure cases (Figure 12)

These changes are highlighted in orange in the revised manuscript. Some newly added figures and tables are temporarily placed at the end of the document to ensure that their numbering remains consistent with our point-by-point responses.

In addition, we have improved the overall clarity of the paper, including clearer definitions of mathematical symbols, enhanced figure readability, and corrected citation errors.

Again, we truly appreciate your thoughtful engagement, and we would gladly welcome any further suggestions for improvement.


Sincerely,

The Authors

---

### Meta-Review · Area_Chair_avjY · 2025-12-30

**Summary:**

The paper initially received three negative and one positive ratings. The concerns are mostly about 1) effectiveness of using and scaling multiple representations, 2) experimental comparisons in terms of performance gains and fairness, e.g., against IT2, 3) overall technical novelty, 4) more analysis, e.g., quality of pseudo labels, long-tail robustness.

**Reviewer Concerns:**

The authors have provided responses in the rebuttal to answer initial concerns from the reviewers. The AC took a close look at the paper, reviews, and the rebuttal. After the rebuttal, the AC finds that some questions are addressed with more experiments, e.g., pseudo-label quality, ensembling result and its efficiency. However, there are still remaining concerns that are not addressed well, e.g., effectiveness of using multiple representations (reviewer J17C, A5B5, 5ZQn, qjPV), performance comparisons against IT2 (reviewer J17C, 5ZQn), and technical novelty (reviewer J17C, qjPV). Especially, the AC agrees with reviewer J17C's feedback during the discussion period about technical contributions of using multiple representations. Considering these major concerns not being addressed well in the rebuttal, the AC agrees with the reviewers' overall feedback and hence recommends the rejection rating.

**Reviewer Scores:**

Reviewer qjPV mentioned to retain the original positive rating, while the other three reviewers did not fully participate in the discussion. For reviewer J17C, during the discussion period, it appears that the main concerns are not fully addressed, so there is no clear evidence that the rating may be increased after rebuttal.

---

### Decision · Program_Chairs · 2026-01-26

Reject